# Traceability Management of Socio-Cyber-Physical Systems Involving Goal and SysML Models †

**Amal Ahmed Anda** , **Daniel Amyot** * **and John Mylopoulos** 

School of Electrical Engineering and Computer Science, University of Ottawa, Ottawa, ON K1N 6N5, Canada; aanda@uottawa.ca (A.A.A.); jm@cs.toronto.edu (J.M.)
* Correspondence: damyot@uottawa.ca; Tel.: +1-613-562-5800 (ext. 6947)
† This paper is an extended version of our paper published in Anda, Amal Ahmed, and Daniel Amyot. Traceability management of GRL and SysML models. In Proceedings of the 12th System Analysis and modelling Conference (SAM), Canada, 19–20 October 2020; pp. 117–126.

**Abstract:** Socio-cyber-physical systems (SCPSs) have emerged as networked heterogeneous systems that incorporate social components (e.g., business processes and social networks) along with physical (e.g., Internet-of-Things devices) and software components. Model-driven techniques for building SCPSs need actor and goal models to capture social concerns, whereas system issues are often addressed with the Systems Modeling Language (SysML). Comprehensive traceability between these types of models is essential to support consistency and completeness checks, change management, and impact analysis. However, traceability management between these complementary views is not well supported across SysML tools, particularly when models evolve because SysML does not provide sophisticated out-of-the-box goal modeling capabilities. In our previous work, we proposed a model-based framework, called CGS4Adaptation, that supports basic traceability by importing goal and SysML models into a leading third-party requirement-management system, namely IBM Rational DOORS. In this paper, we present the framework's traceability management method and its use for automated consistency and completeness checks. Traceability management also includes implicit link detection, thereby, improving the quality of traceability links while better aligning designs with requirements. The method is evaluated using an adaptive SCPS case study involving an IoT-based smart home. The results suggest that the tool-supported method is effective and useful in supporting the traceability management process involving complex goal and SysML models in one environment while saving development time and effort.

**Keywords:** adaptation; consistency; completeness; cyber-physical systems; goal modeling; GRL; SysML; traceability management



## 1. Introduction

*Socio-cyber-physical systems* (SCPSs) consist of networked social, physical, and software subsystems, including business processes, social networks, Internet-of-Things (IoT) devices, and software components. Examples include existing systems, such as ones for air traffic control, and emerging ones, such as smart homes/cities [1], adaptive Systems of Systems (SoS) [2], and intelligent production networks [3]. SCPSs have emerged as diverse, complex systems that raise new challenges for systems engineering. Engineers developing such systems need to consider software, hardware, and stakeholder concerns as well as runtime decisions for SCPSs that need to adapt to their operational environment (i.e., *adaptive* SCPSs).

As reported in a recent literature review [4], model-driven engineering methods, e.g., using the *Systems Modeling Language* (SysML) [5], which supports system specification, and goal-oriented requirements engineering techniques, which support social concepts and strategic decision making, have been used in the recent past to handle the complexity of such systems.

Moreover, the integration of goal models with all system development and evolution activities provides unprecedented opportunities for helping engineers understand and validate system functionality, behavior, and quality in a holistic way [6–8]. Traceability should, hence, be specified between goals, system requirements, design elements, and implementation components [9] as this facilitates and enhances system development and evolution. A comprehensive engineering methodology for building SCPSs should also support consistency and completeness checks, verification, validation, change management, and impact analysis [10,11].

A common approach across the literature is to develop goal models and *remodel* them inside SysML design tools, e.g., by capturing goal models using extensions of requirements or use-case diagrams, and then link them to design elements [12]. In the frequent case where the goal models were created in a different modeling environment that supports goal-oriented analysis, this often causes duplication of work, additional effort and time, and the introduction of inconsistencies [4]. Visualization, filtering, and analysis techniques are essential to help developers track the connected element, detect incomplete and inconsistent elements, and determine the impact of changes. All of these techniques are not well supported by modeling tools. Consequently, to support these activities, existing approaches tend to [4]:

1. Develop a new tool to facilitate and simplify tracking methods by slicing the connected models to create simple and easy-to-follow views. However, developers have to check the consistency and completeness manually using these views, which could provide misleading information if some links are missing or incorrect.
2. Export the models after linking them using requirements or use-case diagrams and importing them into external tools, such as Excel, which also does not provide good support for different types of analyses and visualizations.
3. Provide little guidance or suggestions about (i) how to exploit the created links to assess consistency and completeness across multiple views or models, (ii) the impact of the changes, and (iii) any type of analysis.

For the above reasons, we are interested in developing a comprehensive engineering methodology for SCPSs. As a starting point, we recently proposed a modeling framework called CGS4Adaptation [13–15] that integrates the goal-oriented requirement language, SysML, and feature models, to support through models the development process from requirement analysis to implementation with minimal work duplication or information loss.

SysML reuses a subset of UML with extensions that target systems engineering, e.g., to model designs that combine hardware and software components. SysML includes UML use case, sequence, activity, and state-machine diagrams and is useful for modeling SCPSs. Due to its language features, it simplifies system design [16] and reduces complexity [17] by providing block definition and internal block diagrams from UML class diagrams. In addition, SysML includes parametric and requirement diagrams not included in standard UML to relate a system's design to its requirements [18]. However, the latest SysML standard [5] does not include important first-class concepts for modeling and analyzing social concepts, such as goals, indicators measuring their degree of satisfaction, and contributions that indicate the impact of satisfying a goal on other related goals.

To supplement the concepts found in SysML for capturing social concepts in SCPSs, a complementary goal-oriented modeling language (such as KAOS [19], *i\** [20], or others [21]) can be used. Among these many options, we adopt the *Goal-oriented Requirement Language* (GRL) [22], which enables modeling and analyzing the goals of both a socio-cyber-physical system-to-be and the social actors who participate in its activities. GRL is also part of the *User Requirements Notation* (URN) standard [22]. In addition to design alternatives, GRL supports self-adaptive systems through quantitative and qualitative trade-offs and what-if analysis, which can be conducted inside and outside goal modeling tools.

To support evidence-based decision-making with the best adaptation alternatives according to the current context, GRL uses key performance indicators (KPIs) to monitor

the surrounding environment [22] while propagating their values to measure overall system satisfaction that can be used to adapt accordingly [13,23].

The CGS4Adaptation framework [13] (Figure 1) integrates three modeling languages for the development of SCPSs in a way that enables adapting to user concerns through monitoring quality and compliance:

- SysML, for capturing concerns related to software and hardware.
- GRL, for capturing stakeholder concerns and preferences as well as indicators for measuring them.
- Feature models [24], captured with an extension of GRL, for specifying variability and adaptation opportunities.

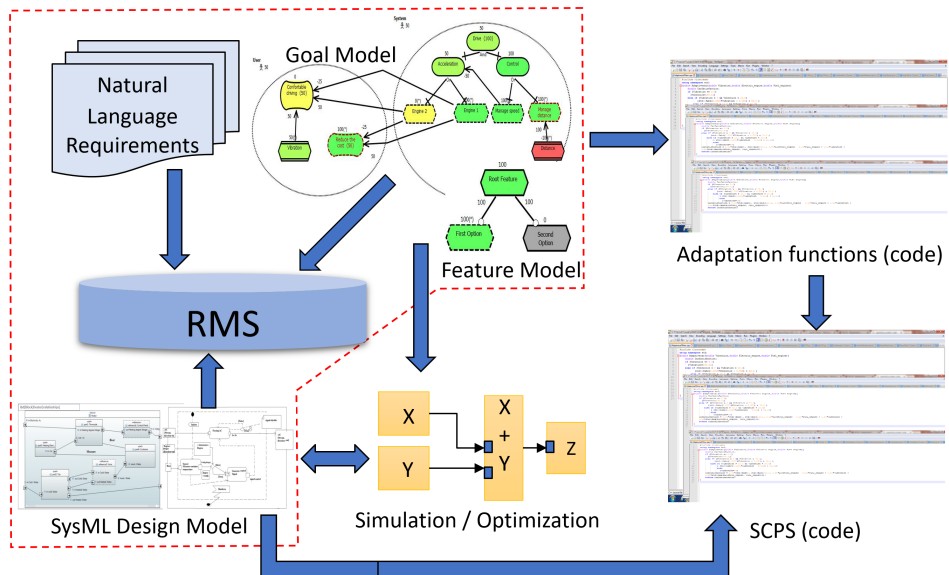

**Figure 1.** CGS4Adaptation overview, with the focus of this paper shown within the dashed line.

The CGS4Adaptation framework consists of two complementary parts supporting traceability and self-adaptation. The first integrates GRL and SysML models using a third-party traceability system (e.g., a requirement-management system) while the second transforms goal and feature models to mathematical functions for effective and efficient goal-based reasoning and adaptation in SysML models (at design time) and in implementations (at runtime) [13,23]. SysML design environments can be used to support simulation, optimization, and code generation of SCPSs, using the functions generated from goal/feature models.

This paper focuses on the area of CGS4Adaptation delimited by the dashed line in Figure 1 and on the RMS-based management and exploitation of traceability between goal models, SysML models, and conventional natural language requirements. An RMS-based solution is meant to help engineers benefit from existing capabilities to manage traceability links, track changes and their impact through these links, and provide sophisticated analysis, visualization, and filtering techniques instead of using design tools that lack (or do not well support) such capabilities. These differences will be further discussed in the related work (Section 3).

In previous work [25], we explored exporting GRL and feature models from the jUCM-Nav [26] tool and SysML models from Cameo Systems Modeler [27] to runnable scripts that IBM Rational DOORS [28,29] runs to import and store the elements of these models and their relationships. DOORS, a popular RMS, can then be used to manage traceability links between the models and to/from conventional requirements. Preliminary completeness and consistency rules are also introduced to support the traceability management process and improve alignment. Completeness is used to assess whether elements in one model are properly addressed (and linked) in another model, and violations at that level may lead to

the addition or deletion of model elements or links. Consistency checks whether elements of appropriate types and natures are linked across models.

In this paper, we extend this previous work by contributing:

1.  A comprehensive and practical model-based traceability and development method to integrate the social concerns modeled by a goal-oriented approach with SysML designs through a requirements management system (RMS).

    -   One particular implementation of this method is provided for the GRL modeling language and for the DOORS RMS. Section 2 provides additional details on why GRL and DOORS were chosen.

2.  Improved and systematic consideration for the goal-based rationale behind the requirements and design elements to facilitate system documentation, traceability, and its applications, including consistency and completeness checks, change management, and impact analysis. This includes:

    (a) The automation of consistency and completeness checks for goal and design models, implemented as executable scripts, accompanied by views and filters to improve the usability of the analysis. The proposed traceability information model (TIM) [11,30] and the consistency and completeness rules are used to implement automatic checks that exploit the direct and indirect relationships between system goals, requirements, and blocks, in order to find alignment issues early and on a continuous basis.

    (b) The definition of new completeness and consistency rules to minimize development time and effort by detecting issues that are difficult to catch otherwise.

    (c) Documentation describing how existing features of RMSs (views and filters) can be used to visualize and solve consistency and completeness issues.

    (d) An evaluation of the efficiency and benefits of the proposed traceability approach using an illustrative IoT-based smart home case study provided by the SM@RT Team [31] at the Informatics Research Institute of Toulouse. A *smart home management system* (SHMS) is an adaptive SCPS supporting elderly stakeholders by performing activities, such as selecting a cost-effective energy provider or controlling heating/cooling and other smart home appliances. SHMSs are usually designed from scratch using available devices and often result in design errors [32].

The proposed approach also takes advantage of existing tools and techniques to minimize coding errors and the effort needed for its implementation.

These contributions are of interest to practitioners and researchers who use model-oriented traceability management, goal-oriented modeling, or SysML modeling as it provides original tool-supported features and a method that combines three complementary modeling aspects and supports automated analysis. Many of these features can also be reused in contexts other than the one used here. The paper is of further interest to researchers as it raises awareness of specific challenges and solutions in developing SCPSs, while identifying other challenges that remain to be addressed.

The rest of the paper is structured as follows. Section 2 presents relevant background technologies, and Section 3 discusses closely-related work. Section 4 explains the proposed traceability management method in CGS4adaptation, including consistency and completeness checks. We evaluate this method using a smart home case study in Section 5. Threats to the validity of this research are detailed in Section 6. Finally, our conclusions and future work are presented in Section 7.

## 2. Background Technologies

This section describes the modeling and RMS tools used in the CGS4Adaptation framework as well as existing technologies for importing models.

## 2.1. Goal Modeling Language and Tool: GRL and jUCMNav

GRL models are composed of *actors* that contain *intentional elements* (goals, softgoals, tasks, and resources) as well as *indicators* converting monitored information into satisfaction levels. Intentional elements can be connected via decomposition *links* (AND, OR, and XOR), contribution links (with weights going from $-100$ to $+100$), and dependencies.

*jUCMNav* is an Eclipse-based editor and analysis tool for the User Requirements Notation, including GRL [26]. Different GRL analysis algorithms can be used to assess goal and actor satisfaction [22]. jUCMNav has an open architecture that enables importing and exporting model information in various formats through custom plug-ins. jUCMNav also supports feature modeling (integrated with goal models) and enables exporting GRL and feature models as mathematical functions whose parameters are the satisfaction levels of the leaves of the models [13,33].

GRL is an excellent candidate for integration with SysML because it supports actors and goals but also indicators for monitoring contexts in a way that enables quantitative analysis and self-adaptation decisions [34]. This is also the only internationally standardized goal modeling language at this time.

## 2.2. SysML Modeling Tool: Cameo Systems Modeler

SysML is supported by many modeling, analysis, and code generation environments [35]. Among them, we find *Cameo Systems Modeler* [27], a model-based systems engineering (MBSE) tool from Dassault Systèmes (formerly No Magic Inc.) based on the MagicDraw Modeling Platform. It supports SysML, including package, requirement, block definition, internal block, activity, use case, and parametric diagrams. This tool was particularly selected for CGS4Adaptation because it can be used to simulate and analyze SysML models, and it can support exporting model information using report templates (described using the *Velocity Template Language*—VTL [36]), a feature that indirectly enables some level of integration with an RMS.

## 2.3. RMS: IBM Rational DOORS

There exist many mature tools, commercial or otherwise, for managing requirements and for exploiting traceability links (see, for example, INCOSE's *SETDB Explore Tools list*, https://www.systemsengineeringtools.com/tools-lists, accessed on 11 January 2023). Very few, however, offer extensible integration interfaces, scriptable automation, and an opportunity to import models, or parts thereof, from other tools (beside preexisting bridges that might be provided). This is why we became interested in DOORS as an RMS for CGS4Automation in general, and for this paper in particular.

IBM Rational's *Dynamic Object Oriented Requirements System* (DOORS) [28] is a leading enterprise-level requirement-management tool, often used by requirements engineers involved in CPS development. DOORS manages extensive information to control system compliance with its requirements and facilitates communication and cooperation between cross-functional teams collaborating on a same project.

In addition to capturing typed requirements and typed links between them (with their attributes), DOORS traces and analyzes changes to requirements, providing several views of their impact on the product as a whole. This paper's method exploits and takes advantage of DOORS to trace system artifacts (goals, requirements, and designs) and their changes across development phases, thus, helping to ensure that the system design is continuously in compliance with its system requirements, goals, and quality requirements. However, in order to manage traceability between modeling artifacts, the latter must be first stored in the DOORS database.

A *DOORS database* contains folders, projects, and modules. A *project* is a type of folder that may hold other folders and modules. An *object* is the core element of the database and represents the concept that we want to trace (e.g., a requirement, a GRL goal, or a SysML block).

Each object has a type and may contain additional predefined and user-defined attributes (e.g., status, author, and priority). In the database, a *formal module* contains the objects in a specific order and forms a particular unit (e.g., a system requirements specification, a software requirements specification, or a test suite) whereas a *link module* holds the traceability links between these objects. Link modules essentially define link types (e.g., "refine", "satisfies", or "tests") according to a traceability information model [11,30]. Link instances, visualized in formal modules with triangular annotations, are typically used for navigation between objects, impact analysis, coverage analysis, change propagation, and document generation.

The *DOORS eXtension Language* (DXL) is a domain-specific language (DSL) used by DOORS to extend its features. DXL is a scripting language used to create, delete, modify, and navigate objects, links, and attributes in the database [37]. DXL scripts can also be created to automate various update and validation activities.

DOORS does not support modeling out of the box. However, our approach uses DOORS as an RMS for storing and linking SysML and GRL models in a way that supports completeness and consistency analysis, in addition to out-of-the-box impact analysis.

### 2.4. Model Import DSL (MI-DSL)

The integration of modeling tools can be performed using technologies, such as the *Open Services Lifecycle Collaboration* (OSLC) standard [38], which manages a loose integration between modeling tools and a RMS with changes synchronization. However, not all tools support OSLC, particularly in the Eclipse world where many modeling tools exist.

The MI-DSL environment represents an alternative to OSLC. MI-DSL is a textual DSL used to specify the desired subset of a modeling language whose model instances need to be traced and managed in DOORS [39]. MI-DSL facilitates model import and synchronization (through re-importing models) by generating a DXL library automatically from a modeling language description. This library can then be used to "execute" models exported to DXL, and import their elements, attributes, and links to DOORS. MI-DSL is supported by an Eclipse-based editor (defined with Xtext [40]) that generates the DXL library automatically (via Xtend transformations [41]). Figure 2 describes the metamodel of this DSL.

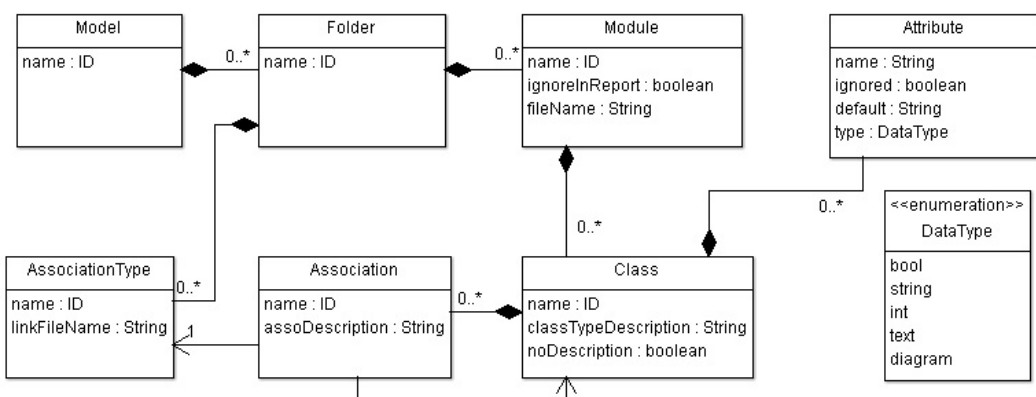

**Figure 2.** MI-DSL metamodel (from [39]).

MI-DSL includes the following concepts:

- *Class*: Describes the modeling language concepts selected for import in the DOORS database. Classes have typed attributes and are linked together through typed associations.
- *Attribute*: Belongs to a class and specifies the name, type, and default value of an attribute.
- *Association*: Belongs to a class and refers to another class. Associations are used to create link sets in DOORS.
- *DataType*: Describes attribute types (Boolean, int, string, text, and diagram).
- *AssociationType*: Describes association types used to create link modules in DOORS.
- *Module*: Is a collection of classes that have similar or compatible attributes.

- *Folder*: Contains modules.
- *Model*: Contains folders. For each generated library, there is exactly one model instance.

MI-DSL is typically used as described in Figure 3. First, starting from a language metamodel (a), the elements, attributes, and links to be tracked in the RMS (in essence, the desired TIM described as an instance of the metamodel in Figure 2) are selected and coded using the MI-DSL editor (b). This is usually performed once per language per project. The MI-DSL editor then generates a DXL library automatically (c) through the use of Xtend-based transformations [39]. The MI-DSL editor and code generator are available at https://github.com/Smart-Contract-Modelling-uOttawa/Traceability/tree/main/MI-DSL (accessed on 11 January 2023).

The tool that supports the modeling language (d) also needs to generate DXL code (e.g., using a plug-in in Eclipse or report templates in VTL), and this is, again, performed once per tool per project. A DXL script can then be generated automatically from each model created by this tool (e). These scripts describe the model elements, attributes, and links of interest, as specified in the MI-DSL description, and invoke the library generated in (c). DOORS can then be used to run such scripts to import a model in its database, or re-import new versions of that model when it evolves (with appropriate warnings).

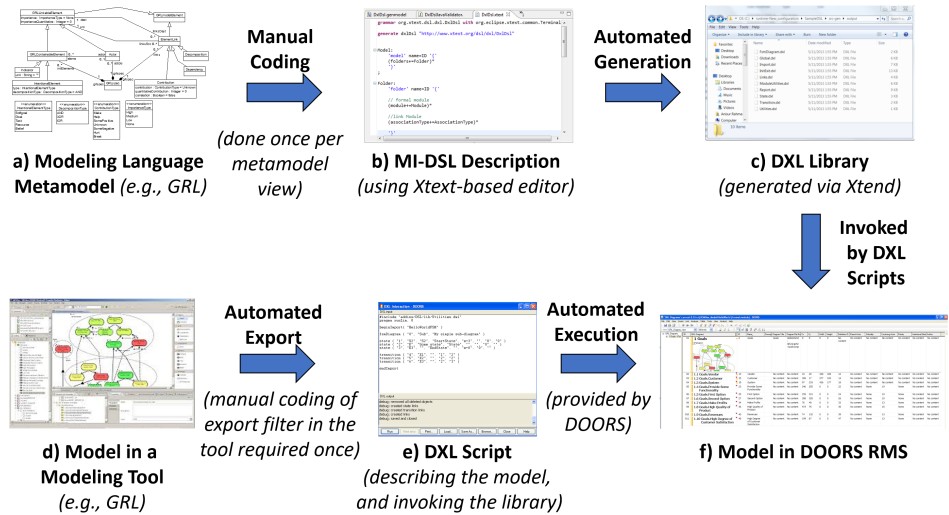

**Figure 3.** MI-DSL process overview.

MI-DSL was used in the past to support importing models in various languages, including Finite State Machine (FSM), the User Requirements Notation (including links between URN's goal and process views), and aspect-oriented URN [39,42–44]. Prior to this paper, MI-DSL had never been used on SysML, on a commercial tool, or in a non-Eclipse-based environment.

In this paper, we evolved an existing plug-in for jUCMNav (a) to export the GRL elements required by our traceability information model (b). Similarly, we also created a new VTL-based export mechanism for Cameo Systems Modeler that generates DXL code describing the SysML elements, attributes, and links of interest. Two DXL libraries (c), one for each tool, enable importing evolving GRL and SysML models in DOORS. We also improved the MI-DSL infrastructure itself along the way to fix several bugs and make it more robust.

## 3. Related Work

The need to integrate goals with a system design has been addressed in the literature in multiple ways to support CPSs development and reuse—for instance, with profiles within SysML or through integration with external languages. A recent literature review by Anda and Amyot [4] discussed the state of the art in combining SysML with goal modeling for designing adaptive systems.

To reach effective strategic business alignment, Cui and Paige [45] integrated goals with SysML requirement diagrams via profiles, with extensions to the models and links for tracing and reasoning about requirements. However, they had simple goal models based on the Business Motivation Model standard [46] and considered neither quantitative values of contribution relationships and alternatives between goals (which enable trade-off analysis) nor indicator parameters (which enable monitoring in adaptation contexts) in their extensions.

Ahmad et al. [47] proposed a method to model and validate self-adaptive systems that uses profiles and stereotypes to extend the SysML requirements model with goals, softgoals, and their relationships. Again, this method does not capture indicators, the importance of intentional elements to actors, or contribution weights found in GRL models. In contrast, Badreddin et al. [48] proposed a new textual language (fSysML) to integrate goals and SysML models using a textual syntax to support CPSs development process. However, their results are still preliminary without support for complex analysis or real traceability management.

To handle security goals and requirements, Maskani et al. [49] modified the CompaSRE approach in a way that expanded the requirements profile with security goals and requirements by adding related goals, stakeholders, risks, and assets as attributes. Similarly, to validate complex systems for self-adaptation using requirements, Lee et al. [50] extended SysML requirement diagrams with security requirements and some goal model elements to ensure they are functionally satisfied by a SysML state diagram that displays the information of system states.

To increase support for requirement reuse and reconfiguration in product lines, Wang [12] proposed a multi-level requirement model by integrating textual goals (with AND/OR decomposition) with SysML use-case diagrams and by using a SysML tool to hierarchically classify the requirements in a requirement diagram. However, although they aimed to more systematically manage requirement traceability and facilitate reuse, they imported requirement data and relationships in different software tools (Excel, a spreadsheet, and Teamcenter—a product line management environment) to maintain requirements and their structure and relationships.

These relationships are represented as checkboxes related to each requirement (in Excel sheets) or as a tree structure (in Teamcenter) to remind engineers of the related requirements. Excel is unfortunately a weak RMS solution in general. Furthermore, their approach does not include all of the relationships found between their goals, which, consequently, leads to ignoring some dependencies and other relationships between requirements across related goals. To reinforce their approach, they recommended that system structure and other system development data should be included in their model, which is what our traceability management method does (among other things).

As relations between requirements and the top-level design aid in facilitating an accurate change impact analysis, Wang et al. [51] proposed a traceability model focusing on the requirements of a command system-of-system. These requirements are characterized by their high number and complex relationships. The requirements are decomposed and hierarchically structured with SysML diagrams as a multi-layer requirement traceability model.

The model starts from strategy-task/goal-activity-capability and is mapped to use case, activity, and requirement diagrams to speed up design model updates according to changing requirements. Similar to some of the previous approaches, the impact of goals and tasks on each other across the structure was not considered in the created models. In addition, important traceability-related activities, such as automated consistency/completeness analysis and reporting are not supported.

In the same manner, given the difficulty in adjusting a system according to changing goals and requirements, Mimura et al. [52] proposed a framework that combines elements of GQM+strategies [53] and SysML requirement diagrams, again with a hierarchical structure. They extended the metamodel of GQM+strategies with new relationships to link goals and strategies in the GQM+strategies model with the root requirements of a SysML

requirement diagram in order to enable consistency management. However, no explanation was provided on how the created links should be exploited to facilitate and simplify models to be reviewed and assessed by engineers.

Nejati et al. [54] proposed a framework that includes a methodology for creating traceability links, a traceability information model, and a model-slicing algorithm. Once the main functions of the system are modeled with a use-case diagram, they are linked to safety requirements. These requirements are modeled in a requirement diagram and linked to the system design using SysML *trace* and *decompose* links. To generate traceability reports and simplify the views used to check the consistency of the created links, the framework uses a tool (SafeSlice) that extracts parts of the created models (*slices*) selected according to specific requirements. However, the extracted slices can provide incomplete/misleading information if some links are missing during the linking process.

To link the system goals and stakeholder goals of CPSs, a common approach across the literature [4] is to *remodel* these goals in SysML design tools without importing them from goal modeling tools. Such a strategy is time-consuming (work duplication), limited (not all goal model elements and relationships are usually included in or supported by design tools), error-prone (a manual process), and not scalable given the complexity of CPSs. In addition, traceability management activities, such as visualization, filtering, impact analysis, as well as consistency and completeness checks, are not well supported by design tools. There are also approaches that use profiles to manage traceability from SysML elements to functional and non-functional requirements and to stakeholders (see for example the work of Haidrar et al. [55,56]), without supporting goals and their relationships explicitly, and often without support for automated consistency and completeness analysis and reporting.

Compared to the related work mentioned above, our traceability management method automatically integrates selected elements of interest from goal models (specified in a conceptually-rich and standardized language, namely GRL) and SysML system designs via an RMS with mature and diverse capabilities, without work duplication or information loss, even as models and other tracked artifacts (natural language requirements, tests, etc.) evolve. In addition, our method exploits imported intra-model relationships to save engineers time and effort when linking goals, requirements, and design components together. Useful and usable insights are provided using automatic consistency and completeness checks powered by the RMS ability to visualize, filter, and track a large number of model elements and their internal and external relationships.

One weakness of this method compared to approaches that only use a SysML modeling environment (or a SysML tool and an RMS) is to rely on the use of three tools (goal modeling environment, SysML environment, and third-party RMS) to support design and analysis activities. Our method risks hindering the overall usability (e.g., because of tool context switching) and productivity (e.g., due to the required model export/import activities). In many projects, this may, however, be a small price to pay compared to the cost of completeness and consistency issues discovered too late.

## 4. Traceability Management Method

In this section, we propose a new traceability management method that aims to minimize common hard-to-manage traceability issues by integrating goal and SysML models using a third-party RMS. As shown in Figure 4, this method has two main parts: (1) *preparation*, which describes the steps required before managing traceability links between the respective models and conducting analysis, and (2) *consistency and completeness checks*, which describes the tool-supported steps we propose to check the consistency and completeness of the overall system across its different models. In the following, each part of the proposed method is described and illustrated.

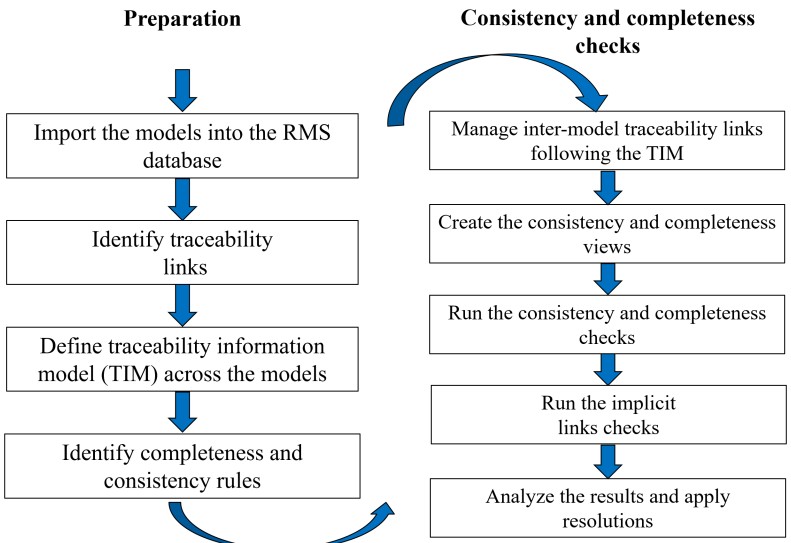

**Figure 4.** Traceability management method.

## 4.1. Preparation

This subsection describes the preparation steps of CGS4Adaptation's traceability management method as shown on the left side of Figure 4.

### 4.1.1. Import the Models into the RMS Database

The elements of the GRL and SysML models, together with system requirements, must first be imported into the RMS repository. Importing mechanisms differ according to the various modeling tools used (e.g., jUCMNav and Eclipse for GRL, and the Cameo Modeling System and VTL for SysML). Since jUCMNav already supports exporting GRL models to DXL scripts that invoke an existing DXL library [39], this paper focuses on describing SysML's modeling elements, attributes, and associations of interest, following the approach described in Section 2 and Figure 3.

Anda and Amyot [25] provided more details on how to describe SysML models using MI-DSL classes, how to generate the DXL scripts, and how to import GRL and SysML models into a DOORS database. Below, we present the main steps of the import process, including defining the elements to track, describing them using MI-DSL, creating the corresponding library, and finally creating DXL scripts from the design tool. The full VTL script for exporting SysML models in DXL, the MI-DSL and DXL library needed to import them in DOORS, and the automated checks are available at https://github.com/Smart-Contract-Modelling-uOttawa/Traceability (accessed on 11 January 2023).

**Selected SysML elements:** SysML is a large and complex language where only a fraction of the language's concepts is of interest here. In order to support relevant types of analysis, avoid duplication of work, and avoid the degradation of the RMS' performance, only two important SysML diagrams, namely requirement diagrams and block definition diagrams (BDDs) as well as their elements are selected for traceability support in the RMS. Table 1 shows the selected elements and relationships from these diagrams, the rationale for choosing them, their roles in the integration, and related attributes in the MI-DSL classes.

Note that the objective here is to check whether requirements and design blocks meet the goals of stakeholders and systems as well as to fill in the gaps that may be caused by misalignment or incompleteness issues. However, the method is not limited to these two types of diagrams. Engineers have the flexibility to choose which diagrams or elements they want to track (in the design model and in the goal model), and then adapt the TIM we provide here (in MI-DSL) the export filters from the modeling tools (to ensure the required elements, attributes, and links specified in the TIM are exported), and the consistency and completeness checks (to better exploit the new TIM links).

**Table 1.** SysML model elements exported to the RMS. Dig = Diagram, FR = Functional Requirement, and NFR = Non-Functional Requirement.

| Diag. | Element or *Relationship* | Rationale | Role | Attributes |
|---|---|---|---|---|
| Requirements Diagram | Requirement | Describes requirements textually | FRs are linked to GRL goals, and NFRs are linked to GRL soft-goals or to related requirements stored in the RMS. | ID, Name, and Text |
| | *Containment, Derive, Satisfy* | Describe the SysML relationships between requirements as well as between requirements and blocks | | ID, Name, Source, and Destination IDs |
| Block Definition Diagram | Block | Core element representing software, hardware, material, or a process | To be linked to GRL goals, soft-goals, indicators, or tasks | ID, Name, and Type |
| | *Part* | Describes the *Composed-of* relationship between blocks | Supports implicit and explicit traceability and minimizes management effort. | ID, Name, Source, and Destination IDs |
| | *Aggregation* | Describes aggregation relationships between blocks | | ID, Name, Source, and Destination IDs |
| | *Dependency* | Describes dependency relationships between blocks | | ID, Name, Source, and Destination IDs |
| | *Reference* | Describes association relationships between blocks | | ID, Name, Source, and Destination IDs |
| | *Generalization* | Describes type-of relationships between blocks | | ID, Name, Source, and Destination IDs |

**MI-DSL description of SysML:** The MI-DSL specification of the SysML block definition and requirement diagram elements of interest is shown in Listing 1. In addition to the attributes used to describe the imported elements, the MI-DSL classes have three default attributes (Name, ID, and Description) that do not explicitly appear in their specification. MI-DSL provides the following mapping between specified elements and the DOORS database structure, objects, and links:

1. Formal modules: hold the instances of the selected elements of block definition and requirement diagrams, including the diagrams themselves, requirement, block, block relationships, and requirements relationships.
2. Classes describe these elements as DOORS objects with attributes.
3. Linksets: sort the links between objects into groups to facilitate the traceability management process. They are specified by the associations in the classes, such as derivedFrom (links between requirements), satisfiedBy (links between requirements and blocks), containedBy (links between requirements or between an individual requirement and its related requirement diagram), partOf, IsSharedBy, Dependency, refersTo, and childOf.
4. Link modules: represent the links between the created instances, including all the related association types: Composition, Aggregation, Association, Generalization, Dependency, Derive, Satisfy, and ContainedBy.

**Listing 1**: MI-DSL classes of the selected SysML diagrams and elements (extract).

```
model SysMLModel{
folder SysML{

// Requirement diagrams
module requirement{
// Default name, id, description are already provided,
// and the description could contain the Text string.
class requirement{
string "ReqID" shows as "ReqID"
}
}

// SysML relationships involving requirements
module requirementlink{
fileName "Requirements Relationships"
```

```
class requirementlink{
string "sourceID" shows as "Source ID"
string "destinationID" shows as "Destination ID"

association derAsso1 : derivedFrom to  "requirement"."requirement" "source ID"
association derAsso2 : derivedFrom to  "requirement"."requirement" "
    destination ID"
}

// Other relationships skipped...
}

// Block definition diagrams
module blockDefinitionDiagram{
// Default name, id, description attributes are already provided.
// The description could contain the documentation of the block at design time
    .
class blockDefinitionDiagram{
diagram "graphFileName" shows as "Diagram File Name"
string "title" shows as "Diagram Title"
}
}

module block{
class block {
string "Type" shows as "block Type"
}
}

// Additional SysML relationships involving blocks only
module blocklink{
fileName "Block Relationships"

class blockToDiagram{
string "sourceID" shows as "Source ID"
string "destinationID" shows as "Destination ID"

association CompAsso1 : containedBy to  "block"."block" "source ID"
association CompAsso2 : containedBy to  "blockDefinitionDiagram"."
    blockDefinitionDiagram" "destination ID"
}

class BlockDependency{
string "sourceID" shows as "Source ID"
string "destinationID" shows as "Destination ID"

association DepAsso1 : Dependency to  "block"."block" "source ID"
association DepAsso2 : Dependency to  "block"."block" "destination ID"
}

// Other relationships skipped...
}

// Other modules skipped...

// Association types
associationType derivedFrom "Derive"
associationType satisfiedBy "Satisfy"
associationType partOf   "Composition"
associationType IsSharedBy "Aggregation"
associationType Dependency "Dependency"
associationType refersTo  "Association"
associationType childOf   "Generalization"
associationType containedBy "ContainedBy"
}
}
```

**SysML library:** The MI-DSL editor is used to generate DXL library files automatically from the MI-DSL traceability model for SysML shown in Listing 1. The resulting library contains 13 DXL files as described in Table 2. The first seven files are created for each generated library while the others include functions to add, modify, and delete objects in the related formal modules using their identifiers (IDs). For example, the Block.dxl file provides functions to add, delete, and modify the objects in the formal block module in the DOORS database. This function is invoked in the DXL scripts every time we need to import a block instance from the SysML model.

**Table 2.** Generated DXL library files for importing SysML models in DOORS.

| File | Description |
|---|---|
| `Utility.dxl` | This file contains the list of import statements to import all other library files |
| `Global.dxl` | This file declares global variables used in all DXL files in the library. |
| `ModulesUtility.dxl` | This file includes the helper DXL functions invoked during the model import process in DOORS. |
| `Links.dxl` | This file contains DXL library code for the links described in the modules. |
| `InitExist.dxl` | This file contains the DXL functions to initialize and finalize the import process (including GUIs). |
| `Import.dxl` | This DXL library file provides the utility methods that would be invoked to start the model import process in |
| `Report.dxl` | This DXL library file contains generated DXL code for creating a report after the import process. |
| `Requirements-Diagram.dxl` | DXL file for module requirementsDiagram |
| `Requirement.dxl` | DXL file for module requirement |
| `Requirementlink.dxl` | DXL file for module requirementlink |
| `BlockDefinition-Diagram.dxl` | DXL file for module blockDefinitionDiagram |
| `Block.dxl` | DXL file for module block |
| `Blocklink.dxl` | DXL file for module blocklink |

**SysML DXL export from Cameo Systems Modeler:** Listing 2 shows an extract of the reusable template used to export SysML models from Cameo Systems Modeler. This template, written in Apache's Velocity Template Language (VTL) [36], can be invoked and ran as a report from Cameo. VTL supports conditionally iterating through the SysML model elements, attributes, and relationships. The code generated is a DXL script that will invoke the library previously generated.

**Listing 2**: VTL report template for producing DXL scripts from SysML models (extract).

```
#set($var1="#")
$var1 include "addins/DSL/lib/Utilities.dxl"
pragma runLim, 0
beginImport("${project.name}")

// Blocks
#foreach($e in $Block)
block("$e.elementID","$e.name","$e.documentation","Block")
#set($varlast= "")
#foreach($Satss in $sorter.sort($report.filterElement($e.clientDependency, ["
    Satisfy"]),"supplier"))
#set($s=$report.getSupplierElement($Satss))
#if($varlast != $s.elementID)
#set($varlast = $s.elementID)
requirementToBlock("$Satss.elementID","SatisfiedBy","","$s.elementID","$e.
    elementID")
#end
#end

#* DependOn relationships*#
#foreach($parttc in $sorter.sort($report.filterElement($e.supplierDependency,
    ["Usage"]),"client"))
#set($c=$report.getClientElement($parttc))
#if($varlast != $c.elementID)
#set($varlast = $c.elementID)
BlockDependency("$parttc.elementID","DependOn","","$c.elementID","$e.elementID
    ")
#end
#end

// Others skipped
endImport
```

**SysML DXL scripts:** Listing 3 shows the structure of DXL scripts that are generated automatically and used to invoke the SysML library to import the models into the DOORS database. The first line is used to include the DXL library that must be stored in the specified path in the DOORS' folder. The two next lines start the import process in the specified folder, while the last line terminates it. The other lines call functions in the generated DXL library and pass the extracted data from the SysML modeling tool as parameters.

For example, the statement `block("122", "Appliance controller", "Controls home appliances", "Block")` invokes the `block` function in the `Block.dxl` file derived from the Block module and the related block class in the MI-DSL specification shown in Listing 1.

As seen in Listing 3, in addition to the three default attributes (id="122", name="Appliance controller", and description="Controls home appliances"), the block function has a "type" attribute (the last parameter) whose values can be either `Block` or `constraintBlock` here.

**Listing 3**: Sample DXL script describing a specific SysML model.

```
#include "addins/DSL/lib/Utilities.dxl"
pragma runLim, 0
beginImport( "FolderName" )
// Call statements for the created functions corresponding to the MI-DSL
    classes and their parameters
blockDefinitionDiagram("d1", "AdaptiveCarBlocks", "Diagram description", "
    graph.png", "SysML Block Definition Diagram Title" )
block("122","Appliance controller", "controls home appliances","Block")
block("123","Enable air conditioner access","GRL indicator", "constraintblock"
    )
blockToDiagram("c4","ContainedBy","","b2","d1")
blockToDiagram("c5","ContainedBy","","b3","d1")
BlockDependency("d6","DependOn","","122","123")
...
endImport
```

### 4.1.2. Identify Traceability Links

In order to support change management and impact analysis, consistency between the involved models needs to be checked [57]. Thus, traceability links are required between the imported models to assess completeness and consistency as well as to detect property violations, particularly after modifying or deleting linked elements when models are updated. Such activities can be automated using an RMS. Goal models provide the rationales behind requirements and system design; however, they also provide possible design alternatives that are useful for systems with socio-technical concerns [57].

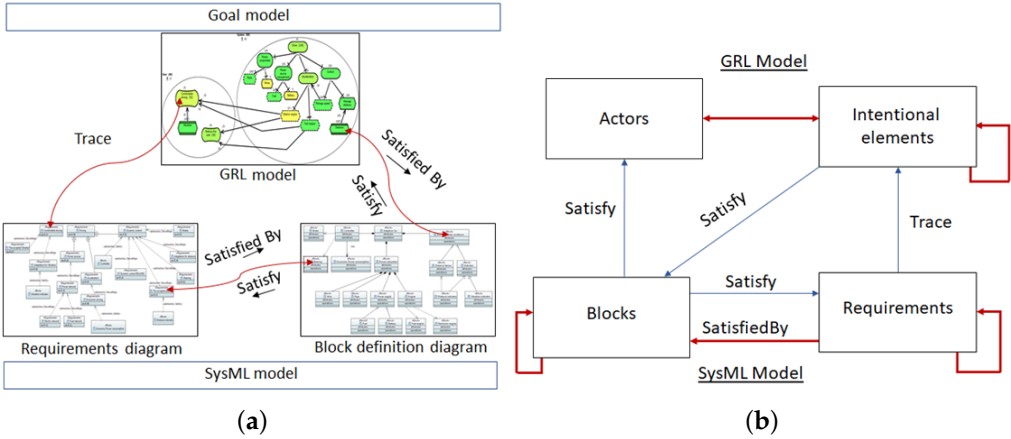

**Figure 5.** (**a**) Overview of traceability in CGS4Adaptation. (**b**) TIM for managing traceability between elements of goal and SysML models in a RMS.

In this context, the initial set of possible traceability links considered in this work (Figure 5a) includes:

1. *Trace* links between goal model elements (intentional elements in GRL) and requirements.
2. *Satisfy* links between design elements (blocks in a SysML BDD) and goal model elements (intentional elements in a GRL model).
3. *Satisfy* links between design elements (blocks in a BDD) and requirements (these links already exist in SysML models).

These links can be set up in both directions. However, not all elements need to be traced. Engineers can select the important elements based on their organizational roles and the nature of the developed project itself [57]. Exploring the benefits of these links and the

potential need for other types of links can support effective SCPS requirements definition, modeling, and analysis.

### 4.1.3. Identify Traceability Information Model (TIM) across the Models

To facilitate tracing the links between models, we need to specify a TIM [11,30], sometimes referred to as a link schema, between the involved models. This TIM/schema formalizes the direction and type of each link connecting two objects in different modules. Figure 5b describes the TIM provided by default in our method, which could be adapted to the specific context of a given project depending on its analysis and reporting needs.

From this TIM, the traceability between the modules is managed by exploiting two types of links: (1) links that are imported from models themselves and represented by red/bold arrows and (2) links that can be added manually by the engineers, represented by blue arrows. The new *Satisfy* and *Trace* links between GRL model elements and SysML model elements in Figure 5b need to be populated manually in the RMS. Other such links (possibly involving other model element types) could be created should the need arise; however, this work only explores these particular links to keep the manual effort to a minimum.

### 4.1.4. Identify Completeness and Consistency Rules

To better manage the traceability between different models, we need to define the consistency and completeness *rules* that exploit the TIM model and can help reveal issues between and within models in the context of SCPS. In our method, such rules include the following:

1. Each GRL element (intentional element or actor) shall be satisfied by one or more blocks.
2. Each block shall satisfy one or more intentional elements.
3. Each intentional element shall be traced to one or more requirements.
4. Each requirement shall trace one or more intentional elements.
5. Each block shall satisfy one or more requirements.
6. Each requirement shall be satisfied by one or more blocks.
7. If the parent/owner of objects is linked to another module, all its associated children/parts shall be implicitly considered linked to the same module (with the same link type).

- In a SysML model, all objects linked to another object in the same module via ContainedBy, PartOf, or ChildOf relationships are considered children of that object.
- In a GRL model, all objects linked to another object in the same module via (AND, OR, or XOR) decomposition relationships are considered parts of that object.

The last rule will help minimize the number of required links to be manually provided. Note that some model elements might be considered *exceptions* to the above rules (e.g., a requirement about the color of a system element may not need to be linked to an intentional element). For enabling modelers to identify false positives explicitly in such cases (particularly when the consistency and completeness analysis is repeated), objects that are annotated with an "ignore" flag will be ignored during analysis, and hence these objects will not violate the rules (see Section 4.2.2).

### 4.2. Consistency and Completeness Checks

To efficiently alignment issues between models and fix them based on analysis results, automated consistency and completeness checks are proposed in our traceability management method. The steps corresponding to these checks, shown on the right side of Figure 4, are detailed here.

### 4.2.1. Manage Inter-Model Traceability Links Following the TIM

Engineers can take advantage of an RMS (e.g., DOORS) to create, manage, and exploit traceability links between the imported models and bridge the gaps between different system views, including goals, requirements, design elements, and possibly others. DOORS

comes out of the box with many impact and traceability analysis features that can be used here.

However, DOORS and other RMSs, by default, do not understand the semantics of the objects being traced, and hence cannot, out of the box, detect inconsistency and incompleteness issues. Such detection mechanisms must be created to exploit *intra-model links* (imported automatically as described in the previous subsection) and *inter-model links* added manually (or possibly semi-automatically using specialized information retrieval techniques, such as those reported by Mäder et al. [58], Rodriguez and Carver [59], Aung et al. [60]). The added links must comply with the types and directions specified in the TIM.

### 4.2.2. Create the Consistency and Completeness Views

In order to keep the results of the consistency and completeness checks usable within the DOORS environment, a new DOORS view called *consistency and completeness* is created for each formal module. In addition to the object names and identifiers, this view shows the following attributes:

1.  Ignore flag: used to determine whether to validate the links of the related object or not.
2.  Consistency and Completeness: describes detected consistency and completeness issues.
3.  Implicit Link: holds the name and identifier of each object that has an indirect link with the current object.
4.  Two linked objects info attributes: each one holds the identifier and name of the object linked to the current object. For example, the Requirements module provides Block Info and Intentional Elements Info attributes, and the Blocks module provides Requirements Info and Intentional Elements Info attributes.

### 4.2.3. Run the Consistency and Completeness Checks

To access the objects of each formal module and check their links, the DOORS DXL is used to analyze this information based on the specified TIM (Figure 5b), and the seven consistency and completeness rules defined in Section 4.1.4. Algorithms 1–4 show the steps of these checks.

The result of these checks are saved in the created consistency and completeness views. The DXL procedure Consistency and completeness checks in Algorithm 1 goes through each object in the identified modules (which can be the Blocks module, Requirements module, Intentional elements module, or Actors module) and checks the out/in links based on the TIM defined in Figure 5b. This procedure searches for the desired links of specific linksets and the related modules (parameters 3) according to the rules identified in Section 4.1.4. It calls two recursive functions, inLink (Algorithm 2) and outLink (Algorithm 3), to search for the incoming and outgoing traceability links in the identified module (parameters 3) via the specified linkset (parameters 2).

In these functions, if the current object (parameters 1) has no link to/from the specified module via the specified linkset, the Parent function (Algorithm 4) is called to check whether the parents have the desired links (Rule 7). Furthermore, the intra-model links imported from the original models are exploited by the Main procedure to cover implicit links and track the parents' chain of objects via the Parent function. Finally, if the inLink and/or outLink functions do not return objects, a consistency and completeness problem is recorded in the corresponding column of the DOORS module being checked. Although not illustrated in the algorithms (to keep them simple), the objects annotated with the "ignore" flag are skipped during the analysis.

---

**Algorithm 1** Consistency and completeness procedure handling rules #1 to #7 in Section 4.1.4.

---

1: *Module* = {*RequirementsModule, IntentionalModule, ActorModule,*
   *BlockModule*} ▷ The exported modules
2: **procedure** CONSISTENCY AND COMPLETENESS CHECKS
3:    **for all** module in Module **do**
4:        set module to Edit mode
5:        **for all** Object in module **do**
6:            Clear the consistency and completeness attributes
7:        **end for**
8:    **end for**
9:    **for all** Object in BlockModule **do** ▷ Rules 2, 5 and 7.
10:       **if** (OUTLINK(Object, Satisfy, RequirementsModule)==**null**
          **and** INLINK(Object, SatisfiedBy, RequirementsRelationshipsModule)==**null**) **then**
11:           Report consistency and completeness violation
12:       **else** Save Object information
13:       **end if**
14:       **if** (INTLINK(Object, Satisfy, IntentionalModule)==**null**) **then**
15:           Report consistency and completeness violation
16:       **else** Save Object information
17:       **end if**
18:    **end for**
19:    **for all** Object in RequirementsModule **do** ▷ Rules 4, 6 and 7.
20:       **if** (INTLINK(Object, Satisfy, BlockModule)==**null**
          **and** OUTLINK(Object, SatisfiedBy, RequirementsRelationshipsModule)==**null**) **then**
21:           Report consistency and completeness violation
22:       **else** Save Object information
23:       **end if**
24:       **if** (OUTLINK(Object, Trace, IntentionalModule)==**null**) **then**
25:           Report consistency and completeness violation
26:       **else** Save Object information
27:       **end if**
28:    **end for**
29:    **for all** Object in IntentionalModule **do** ▷ Rules 1, 3, 7.
30:       **if** (INLINK(Object, Trace, RequirementsModule)==**null**) **then**
31:           Report consistency and completeness violation
32:       **else** Save Object information
33:       **end if**
34:       **if** (OUTLINK(Object, Satisfy, BlockModule)==**null**) **then**
35:           Report consistency and completeness violation
36:       **else** Save Object information
37:       **end if**
38:    **end for**
39:    **for all** Object in ActorModule **do** ▷ Check Rule number 1.
40:       **if** (INLINK(Object, Satisfy, BlockModule)==**null**) **then**
41:           Report consistency and completeness violation
42:       **else** Save Object information
43:       **end if**
44:    **end for**
45:    **for all** module in Module **do** ▷ Implicitly check
46:       **for all** Object in module **do**
47:           Search for implicit links and return the linked Object
48:           **if** (Object != **null**) **then**
49:               Assign the Object to the Implicitly link attribute
50:           **end if**
51:       **end for**
52:    **end for**
53:    **for all** module in Module **do** ▷ Save and close the modules
54:       save module
55:       close module
56:    **end for**
57: **end procedure**

---

---

**Algorithm 2** inLink function: checks specific kind of inlinks related to an Object and its parents.

---

```
 1: function INLINK(Object, Linkset, Module)              ▷ uses the imported/created Links
 2:     if (Object == null) then
 3:         return null
 4:     end if
 5:     if there is link of type Linkset from Module then
 6:         return the source Object
 7:     else
 8:         return inLink(PARENT(Object),Linkset, Module)
 9:     end if
10: end function
```

---

**Algorithm 3** outlink function: checks specific kind of outlinks related to an Object and its parents.

---

```
 1: function OUTLINK(Object, Linkset, Module)             ▷ uses the imported/created Links
 2:     if (Object == null) then
 3:         return null
 4:     end if
 5:     if there is link of type Linkset to Module then
 6:         return the target Object
 7:     else
 8:         return outLink(PARENT(Object),Linkset, Module)
 9:     end if
10: end function
```

---

**Algorithm 4** Parent function: returns the parent of a specific Object.

---

```
 1: function PARENT(Object)                           ▷ uses the imported relationships
 2:     switch Object.type do
 3:         case Block
 4:             if Object is the source of a PartOf/ChildOf link then
 5:                 return the destination Object
 6:             else return null
 7:             end if
 8:         case Requirement
 9:             if Object is the source of a ContainedBy link to a Requirement then
10:                 return the destination Object
11:             else return null
12:             end if
13:         case IntentionalElement
14:             if Object is the source of a Decomposition link then
15:                 return the destination Object
16:             else return null
17:             end if
18:     end switch
19: end function
```

---

4.2.4. Run the Implicit Links Checks

After checking all explicit links for each object, the *implicit links* are explored by the same program to uncover potentially missing links between objects. Here, an implicit link is a relationship that could not be discovered easily by engineers. For example, assume a block is linked to a requirement but not to an intentional element. We check whether the requirement (or one of its parents) is related to an intentional element. If yes, then this block has an implicit link to the same intentional element. The results of this analysis are reported in the Implicit link attribute for each object. Engineers can then filter/flag these new attributes to add potentially missing links.

To save engineers time and effort, these automated checks can be conducted before and after creating new links in DOORS; engineers can efficiently build their traceability links based on the analysis results as explained in the next section.

### 4.2.5. Analyze the Results and Apply Resolutions

When checking the consistency and completeness of the GRL and SysML models, engineers can further explore the results using *DOORS filters* to (1) display the results for specific conditions and (2) apply suitable resolutions in the same consistency and completeness view. For each problematic element, the engineer may then choose one of the following four resolutions:

1.   Add a new Satisfy or Trace link.
2.   Ignore this element during the checks by assigning "True" to its Ignore attribute.
3.   Delete the element from the system design (and such a deletion will be communicated to the other engineers through existing DOORS features).
4.   Tolerate this issue and consider the project as not yet completed.

### 5. Smart Home Illustrative Case Study

We illustrate the feasibility and usefulness of the proposed method via an IoT-based *smart home management system* case study that was first introduced in [25]. This case study is based on the informal description provided by collaborators from the SM@RT Team [31], in France. A *smart home* is composed of in-home services that (i) support elderly and disabled people with the care needed to live independently in the home environment while (ii) providing them with a comfortable and secure life despite age and physical limitations.

We designed goal and SysML models of a self-adaptive SHMS that aims to effectively control most smart devices, such as smart thermostats, smart windows, and home electronics, with the ability to control electricity consumption and production as well as the comfort, privacy, and security of the inhabitants. Smart homes often rely on different IoT devices for monitoring external sources of information as well as for activity recognition inside the home [61]. For simplicity, these are not discussed here, and GRL indicators are used as proxies for such monitored devices. Note that the models and corresponding DXL scripts are available online at https://github.com/Smart-Contract-Modelling-uOttawa/Traceability (accessed 11 January 2023).

#### 5.1. SysML Model

This section presents subsets of the goal and SysML models of the SHMS, including a GRL model created with jUCMNav and SysML block definition and requirement diagrams created with Cameo Systems Modeler.

#### 5.2. Smart Home Requirements

The high-level requirements of the self-adaptive smart home management system are provided in Table 3.

#### 5.3. Smart Home Goal Model

We created a GRL model for SHMS according to the Smart Home case study description from [31] and the requirements from Table 3. The model is divided into several diagrams, which are shown in Figures 6–8. In GRL, actors (dashed ellipses) contain intentional elements, which are mainly goals (rounded-corner rectangles), softgoals (clouds), and tasks (hexagons), together with indicators (hexagons with two lines at the top and bottom). AND/OR/XOR decomposition links are labeled accordingly, contributions are arrows annotated with a numerical positive/negative weight, and dependencies are shown with the filled D links. The small triangles indicate the existence of incoming/outgoing traceability links within the URN model in jUCMNav.

**Table 3.** Requirements of the smart home management system case study from [62]. Ct. = Category.

| Ct. | Id | Description |
|---|---|---|
| Security | SH-01-010 | The Smart Home shall support the prevention and detection of unauthorized physical intrusions. |
| | SH-01-020 | The Smart Home shall support the prevention and detection of unauthorized computer intrusions. |
| | SH-01-050 | The Smart Home shall be able to detect signs of fire. |
| | SH-01-070 | The Smart Home shall allow physical access to emergency services (firemen, hospital service, etc.). |
| Accommodation | SH-02-010 | The Smart Home shall accommodate the various physical, medical, and mental conditions of the inhabitants. |
| | SH-02-020 | The Smart Home shall accommodate specific preferences entered by the inhabitants. |
| | SH-02-030 | The Smart Home shall learn from the behavior of inhabitants. |
| | SH-02-040 | The Smart Home shall assist inhabitants with certain everyday tasks. |
| Economy | SH-03-010 | The Smart Home shall be energy efficient. |
| | SH-03-020 | The Smart Home's annual energy consumption shall be less than the maximum imposed by local regulations. |
| | SH-03-030 | The Smart Home shall support the production and efficient use of energy via solar panels, wind turbines, etc. |

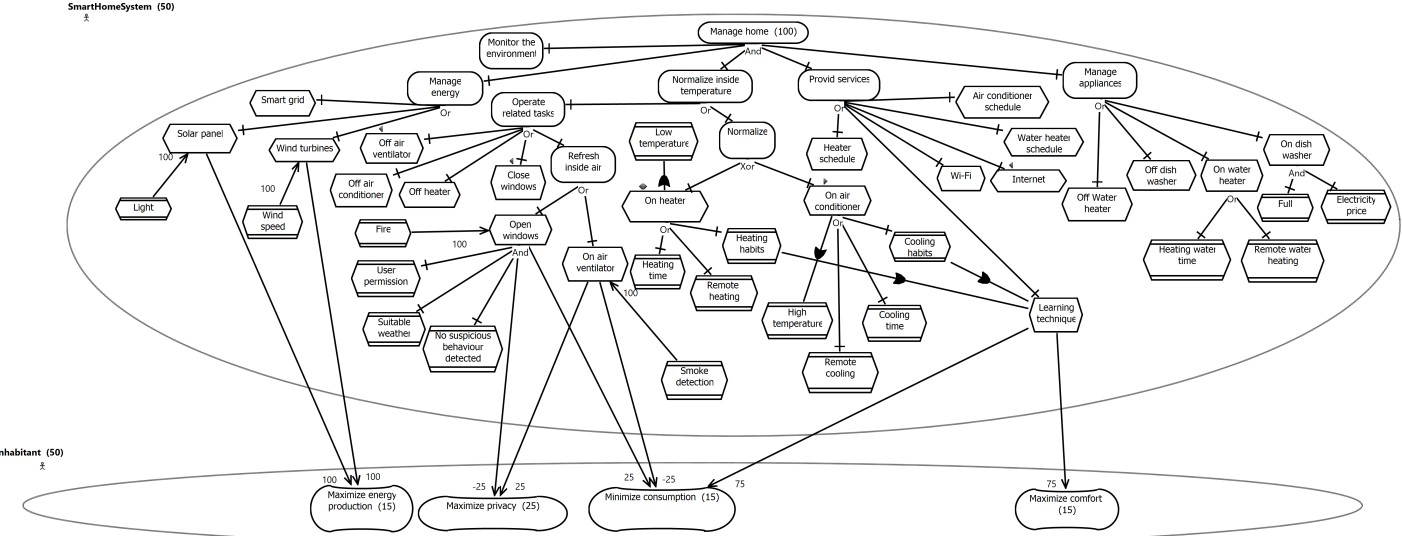

**Figure 6.** GRL model of the top-level Manage home goal.

These three views of the GRL model are sufficient to illustrate the typical content of the goal model of a SCPS without going further into the specifics of the language or of the example, which are outside the scope of this paper. The construction of goal models in general (in GRL and other languages) is further discussed in the surveys of Horkoff et al. [21] and Goncalves et al. [63].

*5.4. SysML Block Definition Diagram*

The structure of the SHMS design is shown in a SysML block definition diagram (Figures 9–11). These figures show the relationships between system blocks, including hierarchy relationships, associations, dependencies, and decompositions, in addition to the system requirements satisfied by these blocks.

Figure 9 illustrates the first level of decomposition of the Smart Home block, while the second level of decomposition is described in Figure 11, where the Monitor block and its related blocks and constraint blocks are introduced. Furthermore, one level of generalization between the Smart device and its specific devices is shown in Figure 10, where the blocks related to the Appliance controller block are described. Again, these three views are sufficient to illustrate how a typical SCPS can be modeled in SysML, and the interested reader is invited to refer to the work of Holt and Perry [64] for general SysML modeling guidelines.

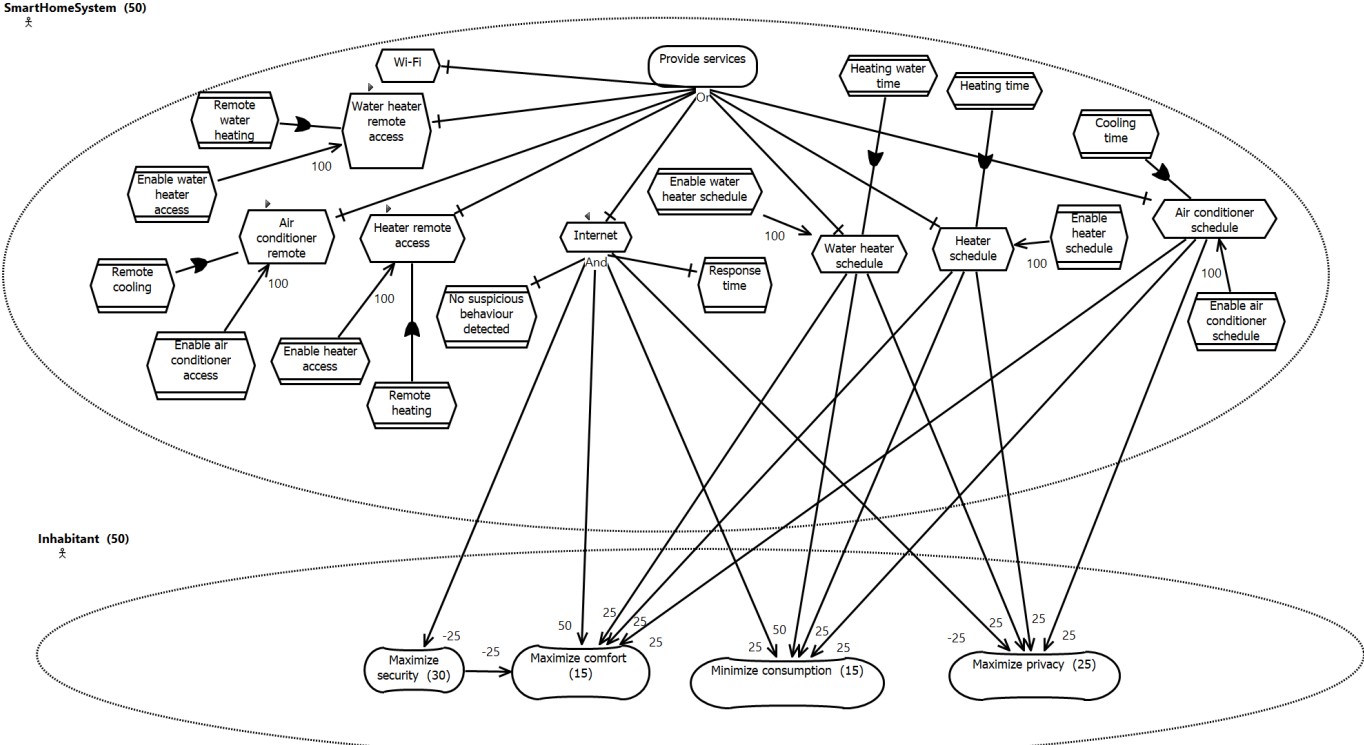

**Figure 7.** GRL model of the Provide services sub-goal.

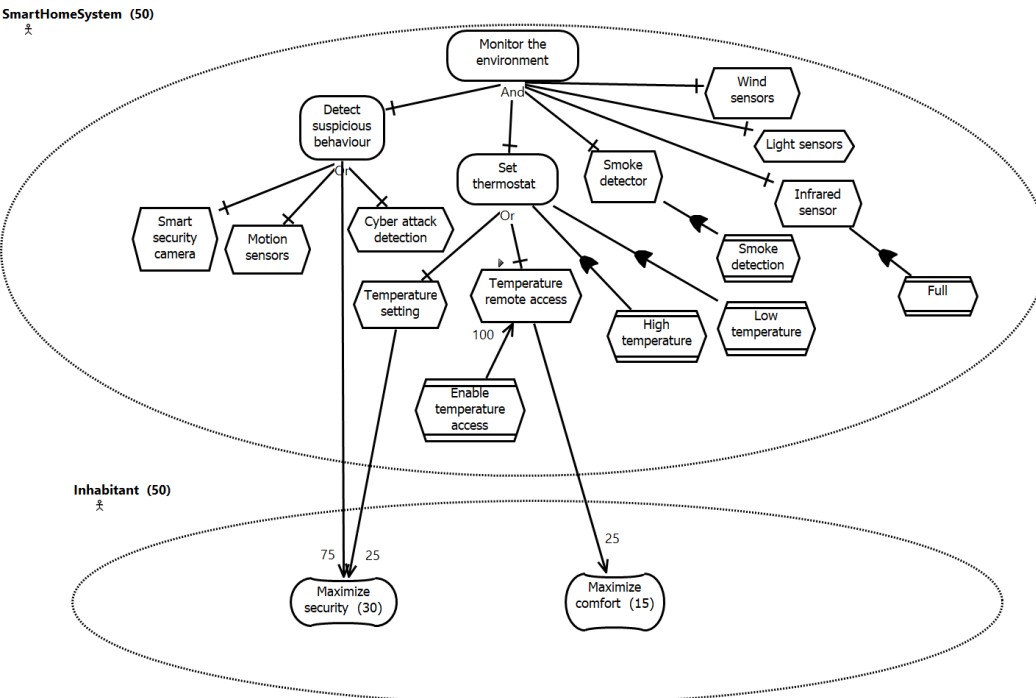

**Figure 8.** GRL model of the Monitor the environment sub-goal.

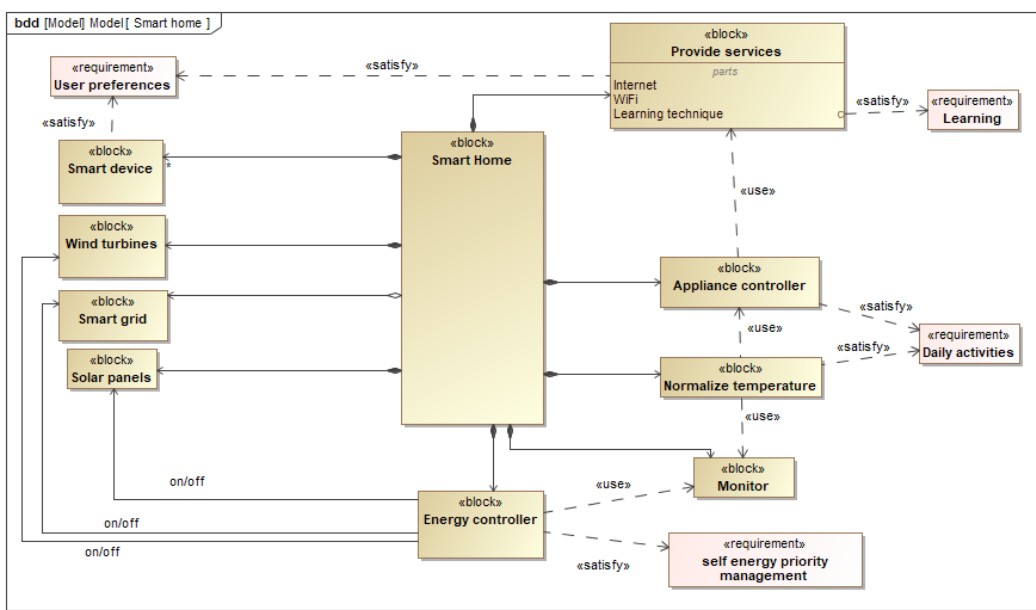

**Figure 9.** Top-level SysML block definition diagram of the SHMS.

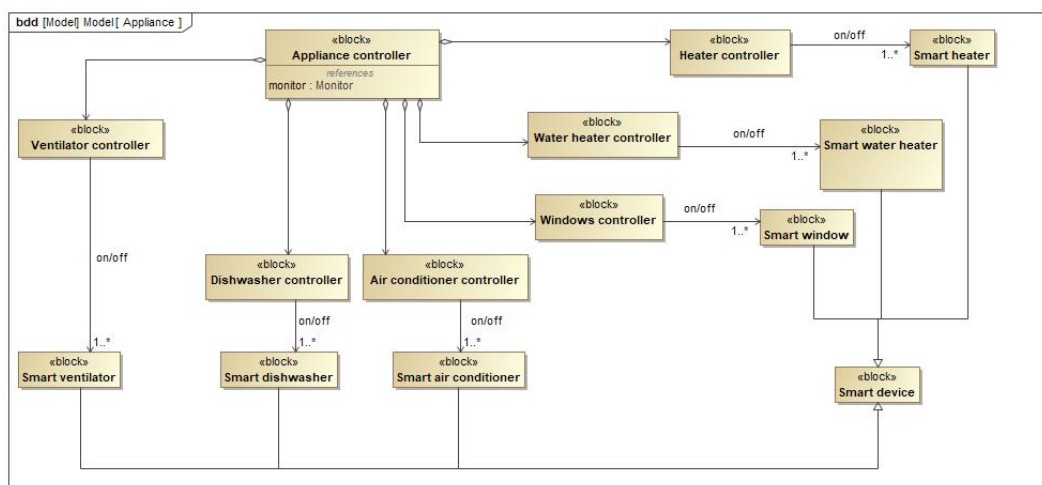

**Figure 10.** SysML block definition diagram of the Appliance controller block from Figure 9.

### 5.5. SysML Requirement Diagram

Figure 12 shows the SysML requirement diagram of the SHMS according to the requirements listed in Table 3. The diagram shows a compound requirement (SmartHomeSystem) and the related sub-requirements (Economy, SecurityAndSafety, and Accommodation), in addition to their own derived and sub-requirements.

### 5.6. Importing the Models into the DOORS Database

We exported GRL models from jUCMNav and SysML models from Cameo Systems Modeler [27] to DXL scripts (invoking our DXL libraries) that were then executed by DOORS. The result is that the elements of the goal and SysML models shown in the previous subsections are formally represented into the DOORS database using formal/link modules, objects, and attributes. Figure 13 shows the resulting formal and link modules of the goal and SysML models in DOORS.

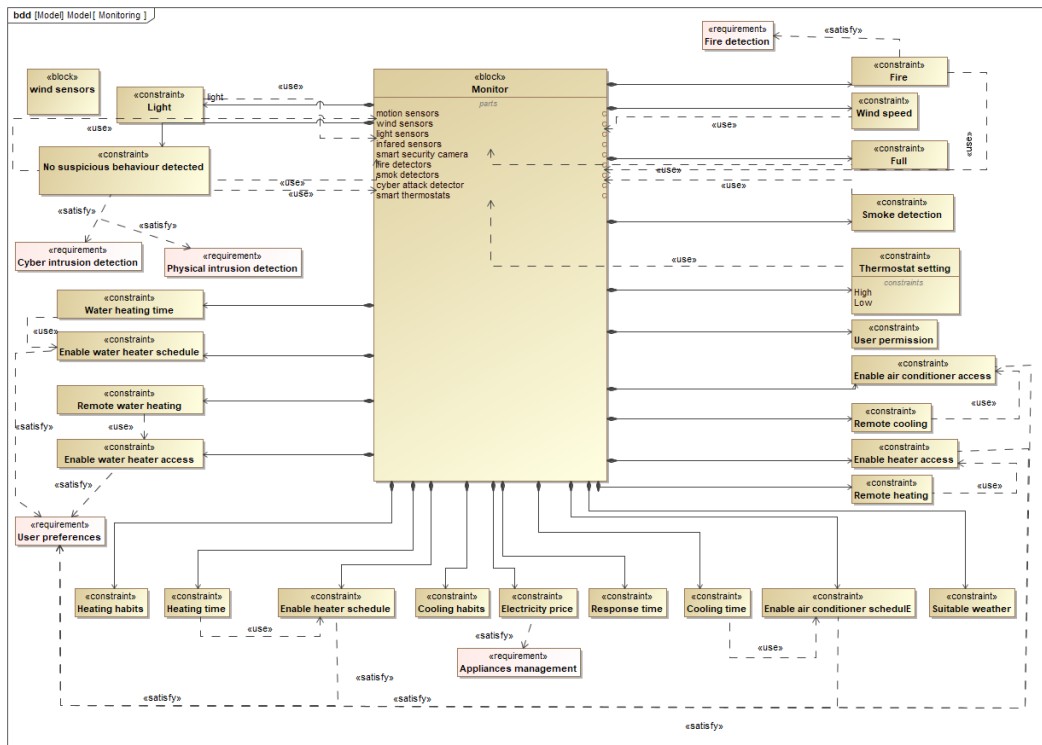

**Figure 11.** SysML block definition diagram of the Monitor block from Figure 9.

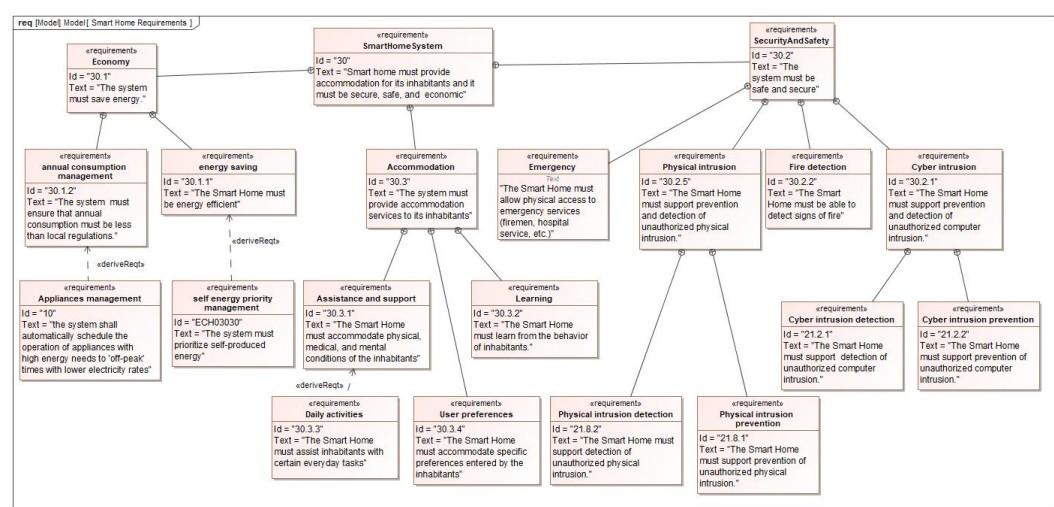

**Figure 12.** SysML requirement diagram of the SHMS.

*5.7. Managing Inter-Model Traceability Links*

We used DOORS to manually create new traceability links between the GRL, requirements, and block definition diagrams following the specified TIM (blue links in Figure 5.b). Furthermore, we created specific DOORS linksets based on the nine link modules imported from GRL and SysML models to satisfy the directions and types of the links specified in the TIM:

1.  A linkset in the Satisfy link module from the Intentional elements module to the Blocks module as shown in Figure 14. In this linkset, we created the following *Satisfy* links between intentional elements and Blocks modules.

    (a)  From the Maximize security softgoal to the Smart security camera block;
    (b)  From the Low-temperature and High-temperature indicators to the Smart thermostat block;

(c)     From the On ventilator and Off ventilator tasks to the Smart ventilator block;

(d)     From the Learning technique task to the Learning technique block;

(e)     From the Maximize comfort softgoal to the Smart device block;

(f)     From the Monitor the environment goal to the Monitor block (but we did not link all their children to the related block/intentional element);

(g)     From the Normalize goal to the Normalize temperature block (but we did not link its related goals and tasks to any block);

(h)     From the Internet task to the Internet block;

(i)     From the Wi-Fi task to the Wi-Fi block (but we did not link the owner block, Provide services, to any intentional element).

2.     A linkset in the Satisfy link module from the Blocks module to the Actors module. We linked the System actor with the Smart home block and we assigned "True" to the ignore attribute of the Inhabitants actor in the Actors formal module (so that it would not create completeness or consistency issues).

3.     A linkset in the Satisfy link module from the Blocks module to the Requirements module. These Satisfy links are created as follows: the Monitor block is linked with the Security and safety and Self-energy priority management requirements while the Assistance and support requirement is linked to the Provide services block. However, there are no links between their children (requirements or blocks) except for the links already imported from the SysML model between these children.

4.     A linkset in the Traced by link module from the Requirements module to the Intentional elements module. In this linkset, the Manage appliances goal is linked to the Appliances management requirement while the Manage energy goal and the Maximize energy production softgoal are linked to the Self-energy priority management requirement. Furthermore, we connected the Minimize consumption softgoal with the Annual consumption management requirement.

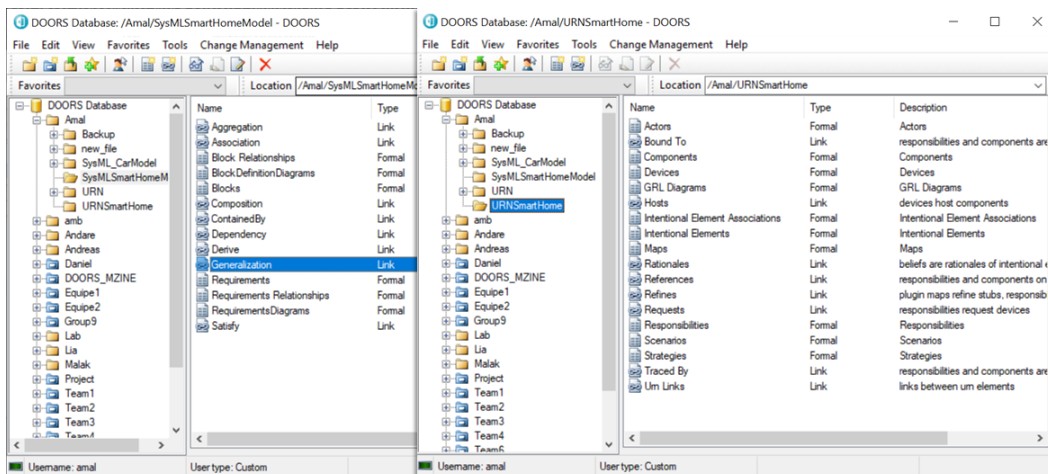

**Figure 13.** SysML model (**left**) and GRL model (**right**) imported in the DOORS database.

Note that we have not connected all the elements together to explain the proposed rules and the benefits of the imported relationships. Creating traceability links could be completed semi-automatically, e.g., using natural language processing [59,60]; however, this is outside the scope of this paper.

*5.8. Results*

Figure 15 shows part of the consistency and completeness view of the Intentional elements module. The Provide services goal is not linked to any block or requirement, as seen in the ConsistencyAndCompleteness attribute, while its children (the WiFi and Internet tasks) are connected explicitly to the WiFi and Internet blocks and implicitly to the User preference requirement. This requirement is satisfied by the Provide services

block, which is the parent of the WiFi and Internet blocks in the Blocks module (Rule #7 in Section 4.1.4).

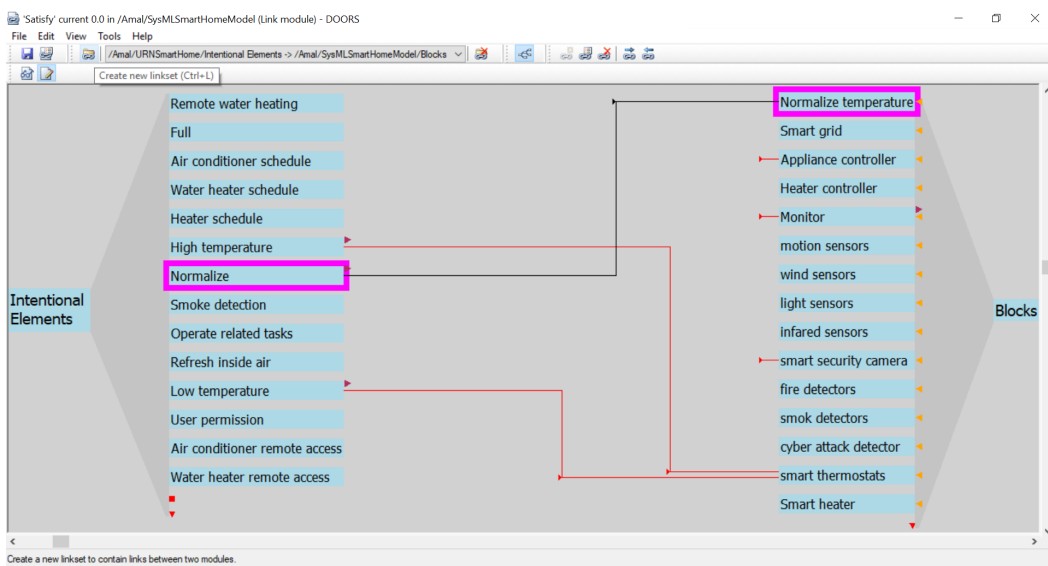

**Figure 14.** *Satisfy* linkset from the Intentional elements module to the Blocks module.

**Figure 15.** Results of the consistency and completeness checks of the Intentional elements module.

This relationship is an implicit one that is imported from the SysML model (see Figure 9) and passed to all parts of the Provide services block during the consistency and completeness checks because these parts do not have any direct link to the requirements model.

Moreover, the Solar panel task does not have any link (there is no triangle next to the object name as shown in Figure 15); however, it is considered to be linked to the Requirements and Blocks modules because its parent, the Manage energy goal, is linked to the Self-energy priority management requirement via a *Trace* link (Section 5.7). Using Rule #7, consistency and completeness checks propagate this relationship to all elements belonging to the Manage energy goal, including Solar panel, Smart grid, and Wind turbines.

Moreover, the implicit link check discovered a *Satisfy* link from the Self-energy priority management requirement to the Energy controller Block (see Figure 9), and so it passed this link to all the intentional elements related to this requirement. As a result, the Solar

panel, Smart grid, and Wind turbines tasks are linked to the Requirements and Intentional elements modules using indirect links that engineers could not easily document otherwise.

Similarity, the Security and safety requirement is linked to the Monitor block, and thus this relationship is passed on to all of its constituent requirements that have no relationships with the Blocks module.

Figure 16 shows the test results where the Emergency and Physical intrusion detection requirements inherit the relationship with the Monitor block from their parent (the Security and safety requirement) while Fire detection does not since it is already connected to the Fire constraint block in the SysML model. In the same manner, implicit links are also discovered through the Monitor the environment goal that was linked to the Monitor block, so these links are passed on to the Emergency and Physical intrusion detection requirements. However, the Fire detection requirement obtained the same implicit link with the Monitor the environment goal but from the Fire block, which is part of the Monitor block. Furthermore, please note that the Appliances management requirement is a *complete* requirement as it does not have any consistency or completeness issues.

**Figure 16.** Results of the consistency and completeness checks of the Requirements module.

From the consistency and completeness view of the Blocks module (Figure 17), the Monitor block is consistent with the Intentional elements and Requirements modules, and its condition is transferred to its parts/children, including Motion sensors and Wind sensors, during the check.

Automating the verification of consistency and completeness rules and of implicit links enables better support for conventional impact analysis and change management analysis by helping to ensure that no link is missing and by showing relevant details about related elements in one usable view.

*5.9. Analyze the Results and Apply Resolutions*

After conducting the consistency and completeness checks, the results are saved in the related views. Some DOORS filters useful for the CGS4Adaptation framework, particularly regarding usability and efficiency, are presented below.

To display design elements that are not linked to goals and requirements, a filter can be applied to the Blocks module to select the objects with empty Intentional elements info and empty Requirements info attributes as illustrated in Figure 18. The result lists 16 blocks and constraint blocks without a rationale (i.e., without a link to a requirement or to a goal). The engineer can then apply the resolution (Section 4.2.5) most suitable for each issue and re-run the checks again.

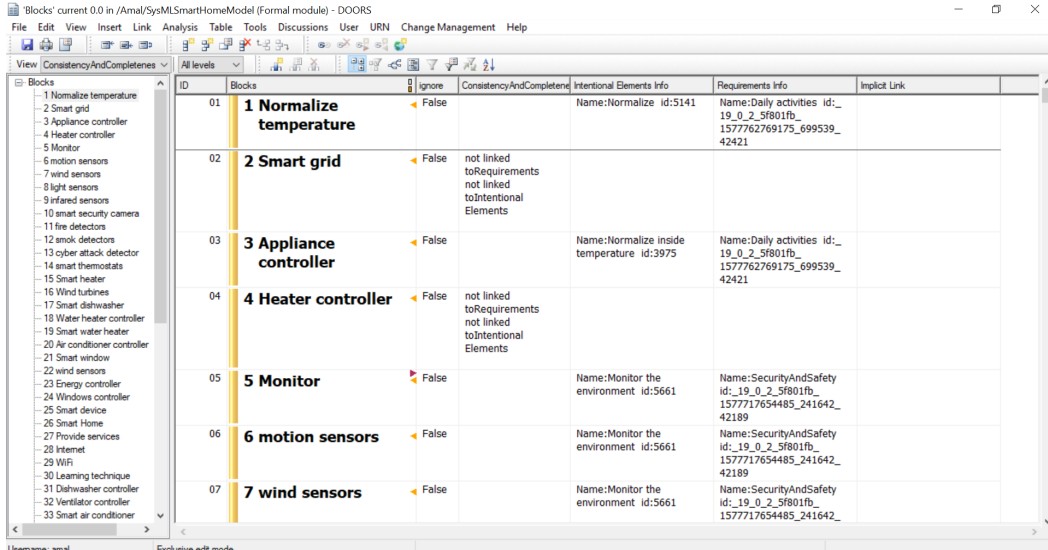

**Figure 17.** Results of the consistency and completeness checks of the Blocks module.

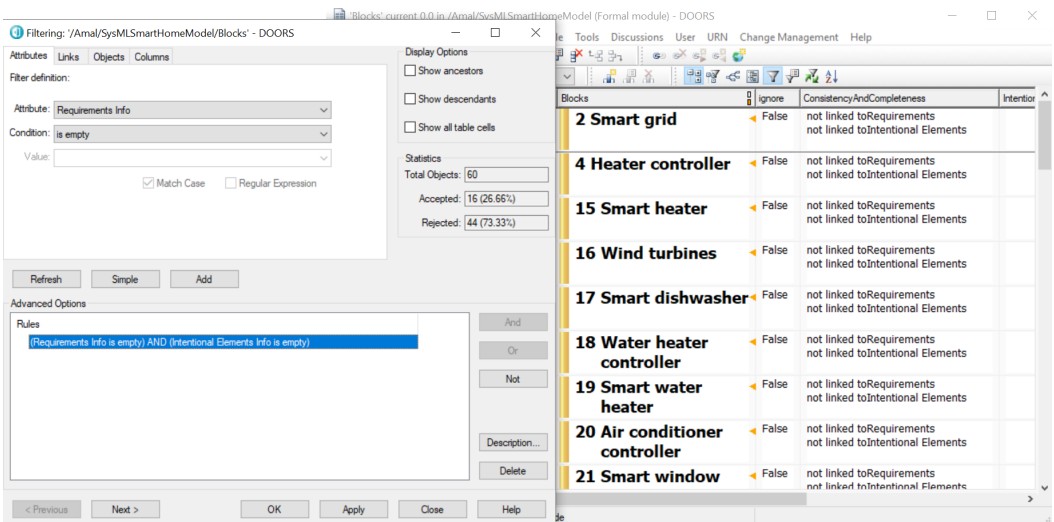

**Figure 18.** DOORS filter: blocks without a rationale.

To extract and review only the satisfied objects (linked to design) and traced objects (linked to requirements) in the Intentional elements module, a filter that extracts objects with a non-empty block info and a non-empty requirements info or a non-empty implicit link, can be applied. The result indicates that 37 elements (softgoals, goals, indicators, or tasks) are complete and consistent in the GRL and SysML models, listed in Figure 19 (right), while there are 39 incomplete or inconsistent elements as displayed in the statistics of Figure 19 (left).

The same filter is applied on the Requirements module to extract all the complete and consistent requirements. The results of this filter, not shown here, include 14 complete requirements (satisfied by blocks and traced to goals) and 6 inconsistent or incomplete requirements.

To list the requirements that have a rationale (linked to the goal model) but that are incomplete (not linked implicitly or explicitly to blocks), a filter that extracts the requirements linked to intentional elements but not to blocks is applied. The filter condition exploits the Implicit Link, Block Info, and Intentional Elements info attributes. The result of applying this filter indicates that the annual consumption management is the only incomplete requirement with a rationale. In this case, engineers can (1) bridge the gap and link this requirement to a related block, (2) ignore this requirement during the checks by

assigning "True" to its Ignore attribute (which is a good option for non-operationalized requirements, such as "The color of the system shall be blue"), (3) delete the requirement, or (4) tolerate this issue and consider the project as not yet completed.

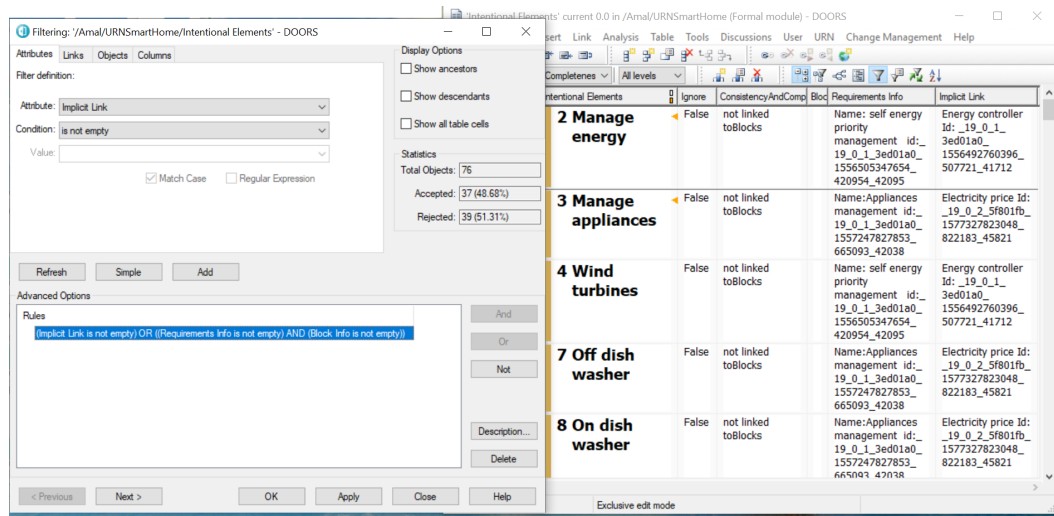

**Figure 19.** DOORS filter: consistent intentional elements.

To list the incomplete requirements without a rationale, a complementary filter is applied. The result (not shown here) indicate four requirements with such issues, namely SmartHomeSystem, Economy, Energy saving, and Accommodation.

### 5.10. Case Study Observations

This case study demonstrated many benefits of our tool-supported traceability management method, including:

- GRL goal models and SysML models (with requirement diagrams and BDDs) of a non-trivial SCPS can effectively be exported from modeling tools via MI-DSL and imported into the DOORS RMS.
- The imported model elements are sufficiently granular to enable important inter-model traceability links to be created according to a TIM.
- The seven consistency and completeness rules defined in Section 4.1.4 and automated through DXL scripts enable the detection of important and non-trivial traceability issues.
- The use of parent–children relationships (inferred from the model's internal links) and of implicit links by the rules reduce the manual effort required to create explicit intra-model links.
- The "ignore" flag (one of the resolutions) enables ignoring model elements that are not meant to be covered by the rules and, hence, avoid many false positives when violations are detected.
- The use of customized DOORS views and filters enables engineers to easily focus on explicit violations and their resolutions.

### 5.11. Performance

This section reports on the performance of many aspects of the method highlighted in Figure 3 and the algorithms of Section 4.2.3. Although we do not provide a formal complexity analysis or a specific experiment targeting performance, we report several important observations. First, regarding the models used in this smart home case study:

- The size of the SysML model is as follows. The Requirements module contains 20 requirements and 58 links related to requirements and other SysML elements. The 3 block definition diagrams contain 60 blocks and 60 embedded links as well as 79 inter-block links.

- The GRL model contains 76 intentional elements, 155 relationships, and 2 actors. The elements and actors are linked to their diagrams via 78 links.

  Regarding Figure 3, where we use MI-DSL and import/export tools:

- *DXL Libraries*: Generating the DXL libraries (part **c** Figure 3, e.g., for GRL and for SysML) from their respective MI-DSL descriptions (part **b**) is fast and takes less than 2 s on a common Windows laptop. However, speed does not really matter here as this step is performed once per TIM.
- *Model Export:* Regarding the generation of DXL scripts (part **e**) from modeling tools (part **d**), Cameo Systems Modeler takes less than 3 s to generate DXL scripts for the SysML model, whereas jUCMNav takes less than 2 s to generate scripts for the GRL model. Both export mechanisms simply navigate a model to generate files, and the complexity is linear with the number of elements and links in the model.
- *Model Import:* To import models (part **f**), DOORS takes longer (but less than a minute) for importing the SysML and GRL models. Checking whether imported elements/links exist and creating, deleting, or updating them is an expensive process in a DOORS database. However, again, this is linear with the number of elements and links.

  Finally, regarding the consistency and completeness checks between the goal and SysML models described by Algorithms 1 to 4:

- A simple analysis of these algorithms suggests that these checks have a time complexity that is linearly proportional to the number of objects in the DOORS modules (the sequence of loops found in Algorithm 1) multiplied by how deep the objects are in their models. This depth results from the traversal of parent–child relationships defined in Algorithms 2 and 3, which use tail recursions (similar to a loop) and are invoked within the loops of Algorithm 1.
- For the case study, DOORS takes several seconds to conduct the complete set of checks, including for explicit and implicit links. The checks go through all the objects; however, not all their links will be explored. Only links belonging to the current object for specific linkSets (i.e., satisfy, partOf, IsSharedBy, Dependency, refersTo, and childOf), and in specific directions are explored according to the introduced TIM.

  Hence, in general, the time complexity is low (linear), and the performance is practical for the types of usage scenarios envisioned for this method.

## 6. Threats to Validity

This section highlights the main threats to the validity of the proposed method and of our research results, including bias and generalization [65].

### 6.1. Construct Validity

Construct validity assesses to what degree the evaluation method measures what the proposed approach claims to do. One threat here is that case studies may not capture the full complexity of real-world problems and models. This is partially mitigated by our case study being based on existing and realistic descriptions and requirements provided by an independent research team. Additionally, we studied the features and some of the values of existing commercial smart home systems in order to create more realistic models.

### 6.2. Internal Validity

Internal validity assesses bias and other factors according to the degree to which conclusions are met based on experimental settings and the collected measurements. One threat here is that having only the first author involved in coding, testing, and reporting the results of the method may have introduced unconscious bias. However, having two more reviewers (the other coauthors) and four external researchers (doctoral thesis examiners) review the goal and SysML models, the rules, the TIM, and the results helped to partially mitigate the risk of bias. Furthermore, the availability of detailed and automatically-generated RMS reports that include the views related to each model, the existing and

additional links connecting the elements, and filtering capabilities helped the authors to verify the results and reduce this risk of bias.

### 6.3. External Validity

External validity assesses the generalization of the results to other areas. The first threat here is that our evaluation was performed via only one case study, and hence the proposed method may not necessarily work as-is in other SCPS contexts or for non-SCPS projects. Another threat is that this method was implemented through specific modeling tools (jUCMNav and Cameo Systems Modeler) and one RMS (DOORS) and has strong dependencies to these tools. This is partially mitigated for the modeling tools by the availability of the MI-DSL technology; however, the latter still heavily depends on DOORS and on a script-based (DXL) execution engine.

The method also relies on many GRL language features and may not be easily adaptable to other goal-oriented languages. As yet another threat, one particular TIM involving GRL and SysML model elements was used in the current incarnation of CGS4Adaptation. Although other TIMs could likely be used for better meeting the particular needs of given SCPS projects, no other TIM was actually explored in this research. Additionally, although our implementation of the proposed consistency and completeness rules depends on that specific TIM, this threat is partially mitigated by the implementation-independent description of these rules.

### 7. Conclusions and Future Work

To support the development of SCPSs (and particularly adaptive ones), connecting social concerns and the design-time and run-time alternatives that satisfy these concerns to system requirements and design is a crucial need. This paper proposed a model-based method to manage the traceability between goal models expressed in GRL and the requirements and design artifacts modeled with SysML. The method supports many activities considered mandatory for SCPS development [66].

This method enables GRL and SysML models to be efficiently created and analyzed in existing tools. By describing the elements of interest in such languages using MI-DSL, desired model elements and their attributes, relationships, and even diagrams can effectively be imported and re-imported (upon model changes) in the DOORS RMS. Using a powerful RMS, such as DOORS, enables sophisticated traceability and impact analyses, not often found in design tools, to be used out of the box.

In order to automate analysis that exploits inter-model links (created manually) and intra-model links (automatically imported), a traceability information model and related completeness and consistency rules were provided. The rules were implemented as executable programs (in DXL for DOORS) that navigate these links and report on inconsistency and incompleteness issues.

We took advantage of DOORS to manage traceability between the imported models while demonstrating how imported and manually-created links can be exploited to automatically check the consistency and completeness of the GRL and SysML models, possibly leading to additional changes in the models, to ensure that they are kept properly aligned. Relevant views and filters were provided to help engineers focus their attention on diverse types of reported issues and on their resolution. We evaluated the feasibility and usefulness of the traceability management method using an IoT-based smart home case study.

The overall results show that importing GRL and SysML models into DOORS to support traceability and alignment is feasible and beneficial. Note that the creation of an MI-DSL description for SysML, the creation of a VTL export script for Cameo Systems Modeler, the generation of DXL libraries and scripts, and the coding in DXL of automated consistency checks took the first authors less than two weeks of development time without any prior experience in MI-DSL, DXL, or VTL (we, hence, expect this to be faster for people with experience in such languages).

Moreover, the results suggest that rules can be checked through explicit links but also through implicit links, often resulting from transitive relationships, hence, detecting issues that cannot easily be spotted by engineers in large and complex models. The use of DOORS views and filters supports the usable management of the traceability information as well as the resolutions (linking, ignoring, or deleting elements or even tolerating an inconsistency) that must be made by engineers. Clean, consistent, and complete links further facilitate common change management and impact analysis processes to likely save engineers time and effort.

Note that these contributions do not all require the simultaneous existence of GRL together with SysML block definition and requirement diagrams. Within SysML, subsets different from the one used here could be involved. For example, requirement diagrams could be skipped if textual requirements exist elsewhere (as they would be redundant). This flexibility provides more opportunities for the above contributions to be used outside the SCPS context targeted by CGS4Adaptation.

The following research directions could be explored for further improvement and increased validity.

- Further evaluation of the CGS4Adaptation framework and its new traceability management method could lead to additional improvements, particularly in terms of generality (e.g., regarding other traceability information models and application areas) and efficiency. This could include industrial case studies and usability studies. An application of CGS4Adaptation that would cover the entire development, implementation, and deployment cycles of a real-world self-adaptive system would be particularly relevant.

- Exploring opportunities to bring GRL/feature modeling concepts and techniques directly into future versions of the SysML standard, in order to minimize the number of tools needed to support adaptive SCPS approaches, such as GS4Adaptation. The upcoming version 2.0 of SysML, whose work started in 2017 [67], will likely include concepts for supporting concerns (akin to goals) and stakeholders [68] but without specific relationships or indicators (which make the richness of a language, such as GRL).

- MI-DSL provides some level of independence regarding the goal and SysML modeling tools used by our method. However, minimizing the dependency to a specific RMS (DOORS) remains an important topic that deserves much attention.

- In a context where the traceability needs of a long-term project will evolve [69], the TIM will likely need some modifications as well, which is a challenge [70]. Providing mechanisms to generate consistency and completeness rules, views, and filters according to a custom-made or evolving TIM involving the goal and SysML languages would also bring important benefits in supporting the needs of projects.

**Author Contributions:** Conceptualization, A.A.A. and D.A.; methodology, A.A.A. and D.A.; software, A.A.A.; validation, A.A.A.; analysis, A.A.A. and D.A.; investigation, A.A.A. and D.A.; resources, A.A.A. and D.A.; data curation, A.A.A.; writing—original draft preparation, A.A.A.; writing—review and editing, D.A. and J.M; visualization, A.A.A.; supervision, D.A. and J.M.; project administration, D.A.; funding acquisition, A.A.A. and J.M. All authors have read and agreed to the published version of the manuscript.

**Funding:** Amal Ahmed Anda was supported by a scholarship from the Libyan Ministry of Education as well as funding from John Mylopoulps's NSERC Discovery grant.

**Institutional Review Board Statement:** Not applicable.

**Informed Consent Statement:** Not applicable.

**Data Availability Statement:** Not applicable.

**Acknowledgments:** The authors thank Dassault Systèmes and IBM for supplying free educational licenses of Cameo Systems Modeler and IBM Rational DOORS, respectively. They are also thankful to B. Jiang, S. Ghanavati, G. Mussbacher, A. Rahman, and X. Zhao for their previous work on MI-DSL,

to J.-M. Bruel for providing the smart home case study, and to J. Sincennes and C. Sibbald for tool support over the years.

**Conflicts of Interest:** The authors declare no conflict of interest.

## Abbreviations

The following abbreviations are used in this manuscript:

| | |
|---|---|
| BDD | Block Definition Diagram |
| CGS4Adaptation | Combining Goals and SysML for Adaptive SCPSs |
| CPS | Cyber-Physical System |
| DOORS | Dynamic Object Oriented Requirements System |
| DSL | Domain Specific Language |
| DXL | DOORS eXtension Language |
| GQM | Goal Quality Metric |
| GRL | Goal-Oriented Requirement Language |
| KPI | Key Performance Indicator |
| IoT | Internet of Things |
| MI-DSL | Model Import DSL |
| RMS | Requirements Management System |
| SCPS | Socio-Cyber-Physical System |
| SHMS | Smart Home Management System |
| SysML | Systems Modeling Language |
| TIM | Traceability Information Model |
| URN | User Requirements Notation |
| VTL | Velocity Template Language |

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
