# Peer review of "Traceability Management of Socio-Cyber-Physical Systems Involving Goal and SysML Models†"

_2673-3951, doi:10.3390/modelling4020009_

Round 1

Reviewer 1 Report

This paper is relevant and takes an interesting approach to traceability management. I thought the experiment is well documented and explains the approach sufficient enough to follow. In line 4 of abstract, seems to be a run-on sentence or incomplete sentence. Suggest reviewing that sentence and correcting. A big question as I am reading this is why not use the SysML use case model since your emphasis is on using SysML and in capturing goals. Perhaps I am missing the uniqueness in the approach that makes the use of Use Case Diagrams insufficient for traceability management.

Author Response

Please see attached PDF file with our answers to all reviewers. Many thanks.

Reviewer 2 Report

Paper reads more like a project description than an academic paper. Not cogent. There appears to be a research contribution here, but the signal to noise ratio is too low to be sure. Paper needs substantial revision such that (i) the problem (traceability) is clearly defined in the context of the relevant literature, (ii) why the approach presented is the right one for this problem, and (iii) constraints and limitations on the approach in resolution of the problem. The problem and the resolution approach must be defined in a tool agnostic manner, although it is perfectly acceptable to illustrate the approach using a specific toolset. As written, the traceability problem is hopelessly confounded with the tools employed.

Author Response

(The authors gave the same response as above.)

Reviewer 3 Report

The performance of the proposed traceability management  method should be compared with the existing methods.

The time complexity of the proposed shold be mentioned.

The content of paper may be organized in a better way.

Author Response

(The authors gave the same response as above.)

Reviewer 4 Report

The article deals with the topic of traceability management in context of Socio-Cyber-Physical Systems. The methods presented are within the model-driven engineering area, to mention Goal and SysML Models. The authors describe their solution bridging these models, as they are not sufficient alone for SCPSs modelling needs. Description include the architecture, algorithms, and implementation details. The solution is verified on a smart house case study and discussed in context of solutions of other authors.

I find this work being of high quality. It is written with the great care of details, thus I do not have any remarks and requests.

Author Response

(The authors gave the same response as above.)

Round 2

Reviewer 2 Report

Significant improvement for this revision. However -- some comments from the first version were not addressed. Furthermore, the review of relevant literature remains at the end of the paper rather than being used up front to provide the context for the research performed by clearly identifying the novel contributions of this research. The paper as presented is not coherent in that the novel contributions to the relevant body of knowledge are not evident. That said, the elements are there and this paper, with significant revision, can be made publishable.

Round 3

Reviewer 2 Report

Now that the paper is organized properly, the flow is much easier to follow. The signal-to-noise ratio on this paper remains low, and the word count could be reduced 25-33% with no loss of content. Serious editing needed to meet the standard typical for an academic paper. No further comments on content.

Author Response

Dear Reviewer,

Thank you for this opportunity to further improve our submission. We have:
- Shortened the introduction and the background to get to the core of the paper sooner;
- Removed a couple of examples in Section 5 that were, to some extent, redundant;
- Shortened several paragraphs and sentences throughout the paper, including in a few tables;
- Reduced the number of figures from 22 to 19.
- Made several editorial improvements.

We were hence able to cut over 2 pages of material without impacting the quality of the paper or the core content. Without precise instructions on what to reduce from Reviewer 3 or the editor, we feel this is the best we can do without impacting what the other 3 reviewers already liked in our paper or requested us to add in previous iterations.